# EX-GRAPH: A PIONEERING DATASET BRIDGING ETHEREUM AND X

**Qian Wang, Zhen Zhang,** [*] **Zemin Liu, Shengliang Lu, Bingqiao Luo, Bingsheng He** [*]
National University of Singapore
`{qiansoc, zhen, zeminliu, lusl}@nus.edu.sg,`
`luo.bingqiao@u.nus.edu, hebs@comp.nus.edu.sg`

## ABSTRACT

While numerous public blockchain datasets are available, their utility is constrained by an exclusive focus on blockchain data. This constraint limits the incorporation of relevant social network data into blockchain analysis, thereby diminishing the breadth and depth of insight that can be derived. To address the above limitation, we introduce EX-Graph, a novel dataset that authentically links Ethereum and X, marking the first and largest dataset of its kind. EX-Graph combines Ethereum transaction records (2 million nodes and 30 million edges) and X following data (1 million nodes and 3 million edges), bonding 30,667 Ethereum addresses with verified X accounts sourced from OpenSea. Detailed statistical analysis on EX-Graph highlights the structural differences between X-matched and non-X-matched Ethereum addresses. Extensive experiments, including Ethereum link prediction, wash-trading Ethereum addresses detection, and X-Ethereum matching link prediction, emphasize the significant role of X data in enhancing Ethereum analysis. EX-Graph is available at `https://exgraph.deno.dev/`.

## 1 INTRODUCTION

Blockchain technology is known for its secure, unchangeable, and anonymous records (Gao et al., 2018). Among various blockchains, Ethereum is one of the largest and has shown substantial growth. The market value of its cryptocurrency, ETH, hit 190 billion dollars as of September 23, 2023 [1]. Another example is Non-Fungible Tokens (NFTs), deployed on Ethereum, attracting numerous celebrities to the blockchain (Nadini et al., 2021). However, the anonymity inherent to blockchain has drawn various financial crimes such as phishing scams and wash trading, particularly within Ethereum, making research on Ethereum necessary (Dyson et al., 2019; Li et al., 2020). Several studies released public Ethereum datasets to facilitate research in this area (Lin et al., 2022; Shamsi et al., 2022). In analyzing activities within Ethereum, researchers are increasingly employing graph analysis, specifically Graph Neural Network (GNN) methods (Liu et al., 2022; Wang et al., 2022).

However, prior studies have failed to address a notable issue: anonymous Ethereum addresses are hash strings which do not contain features outside of blockchain space. This absence of features limits the efficacy of GNN models. Two methods exist to get node features: manually designed features and network representation learning features (Li et al., 2022). However, both types are derived from the Ethereum graph's structure. Recent research indicated that the interplay between features and the the graph structure is a key factor in the deterioration of GNN performance (Yang et al., 2022). We note that some X users utilize the Ethereum Name Service (ENS) for their usernames, associating a human-readable string with a specific Ethereum address. This connection indicates that the Ethereum address and the X account belong to the same entity. One work (Béres et al., 2021) enhanced Ethereum address features by linking 890 X users with ENS-format usernames.

Nevertheless, the reliability of such linkage is questionable as X users can use ENS-like names without any verification. Drawing inspiration from a study (Vasan et al., 2022) which assembled an off-chain network of 14,706 NFT artists along with their verified Ethereum addresses and X data,

---

[*]Corresponding author
[1]`https://coinmarketcap.com/currencies/ethereum/.`

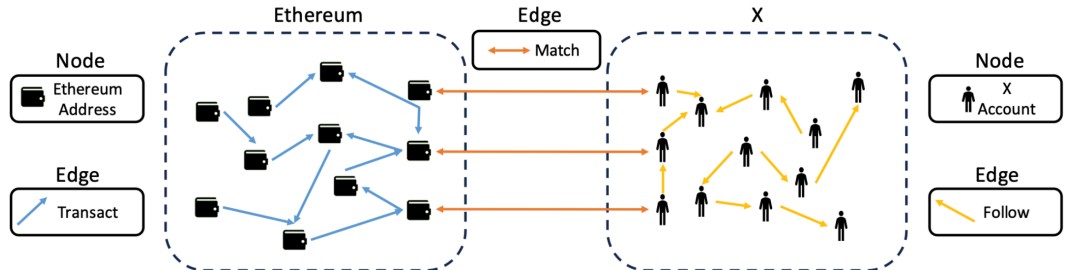

Figure 1: An overview of EX-Graph. The left figure displays the Ethereum transaction graph, showcasing transaction relationships between Ethereum addresses. The right X graph portrays the follow relationships among X accounts. A matching link identifies that the corresponding Ethereum address and X account belong to the same entity.

we consider integrating both off-chain and on-chain networks. This combination aims to enrich the analysis of on-chain activities by incorporating off-chain features. This approach also offers insights for enhancing graph learning on real-world datasets which often lack sufficient features (Huo et al., 2023): integrating features of the same entity from other domains via verified matching links.

To address the limitations of previous research, we introduce a dataset EX-Graph that bridges Ethereum transactions and X follower networks. Our choice of X as an off-chain data source is due to its popularity among blockchain enthusiasts (Casale-Brunet et al., 2022), its accessible public API [2], and the significant impact of tweets on Ethereum NFT market trends (Sawhney et al., 2022; Luo et al., 2022; Tengland & Ask, 2022). Additionally, we collect 30,667 links between Ethereum addresses and their corresponding X accounts from OpenSea [3], ensuring the authentic ownership of each account. Furthermore, we collect 1,445 labeled Ethereum addresses known to participate in wash trading activities from Dune [4] to facilitate Ethereum wash-trading addresses detection. As depicted in Figure 1, EX-Graph consists of approximately 3 million nodes, 33 million edges, and 30,667 matching links that bridge Ethereum addresses with corresponding X accounts.

We further conduct a comprehensive analysis of EX-Graph employing both statistical and empirical methods. Our statistical analysis reveals a notable difference in Ethereum activities between addresses, whether they are matched with X accounts or labeled for wash trading. We observe three key findings from empirical methods. First, integrating X boosts Ethereum link prediction by a peak of 8% AUC (Area Under the Curve) compared to the baseline without X features. Second, the inclusion of X significantly enhances the detection of wash-trading Ethereum addresses, elevating recall by up to 18% relative to the baseline excluding X features. Finally, integrating X supports the innovative task of predicting matching links between Ethereum addresses and their matching X accounts, achieving a 74% AUC. Therefore, EX-Graph provides a valuable resource for future Ethereum-focused research. Our contributions can be summarized as follows:

- We introduce EX-Graph, the most innovative and extensive open-source dataset bridging Ethereum transaction records and X follower network up to date. It lays the groundwork for deanonymizing on-chain activities by utilizing authentic relationships in off-chain contexts.

- EX-Graph consists of the Ethereum Graph and the X Graph, unified by verified matching links. This inclusive dataset captures not only financial transactions on the blockchain but also social interactions about Ethereum on X, providing a more holistic view of Ethereum.

- Through detailed analyses and experiments on EX-Graph, we demonstrate that the integration of X data via verified matching links markedly enhances Ethereum task performances. EX-Graph not only facilitates blockchain research but also contributes to advancements in graph learning, uncovering opportunities for integrating features of the same entity across various domains.

---

[2] https://developer.X.com/en/developer-terms/agreement-and-policy
[3] https://opensea.io/
[4] https://dune.com/queries/364051.

## 2 RELATED WORK

In this section, we review NFT and social media, existing on-chain and on-chain/off-chain matching datasets. We only provide the most related studies here, and leave the rest to Appendix B.

**NFT and social media.** Non-Fungible Tokens, or NFTs, are products on the blockchain which captures much attention. They use blockchain to convert everything possible to digital files such as images, music, and videos (Kapoor et al., 2022). OpenSea is the largest NFT marketplace, which reached total market value of 31 billion dollars volume (Luo et al., 2022). As social media become a place for NFT holders and investors to attract public attention on NFT projects (von Wachter et al., 2022), some works researched on social media such as X, to analyze the relationship between the social media's contents and NFT market. Recent work examined the impact of Tweets on the NFT market through Tweet text embedding (Sawhney et al., 2022), keyword analysis (Luo et al., 2022), frequency of Tweets (Tengland & Ask, 2022), and username analysis (Kapoor et al., 2022). Results of them showed that social media activity, especially on platforms like X, can indeed provide valuable insights for predicting trends and behaviors in the NFT market.

**On-chain datasets.** Many works proposed Ethereum datasets to promote research on Ethereum, focusing on Ethereum link prediction (Lin et al., 2020; 2022; Wang et al., 2022), Ethereum account classification (Huang et al., 2022; Zhang et al., 2024), and Ethereum phishing accounts detection (Chen et al., 2020; Li et al., 2022). In Ethereum transaction graphs, link prediction is to predict whether there will be transactions between two Ethereum addresses. Phishing accounts detection is to detect which Ethereum addresses participate in phishing activities. Chartalist is the first benchmark dataset for machine learning tasks on the blockchain (Shamsi et al., 2022). However, these datasets rely solely on Ethereum, which inherently contains few node features.

**On-chain/off-chain matching datasets.** Few works developed datasets matching on-chain and off-chain activities from same entities. We find two datasets containing matching relationships. One work collected 890 ENS names from X handles or profiles and mapped them to Ethereum addresses (Béres et al., 2021). This methodology, however, lacks reliability since X users can freely alter their displayed usernames. Another limitation of this dataset is its exclusive focus on the Ethereum transaction graph, neglecting the X graph. The other work collected data from 13,487 NFT artists' X accounts, profiles, and Ethereum addresses (Vasan et al., 2022). The limitation of this dataset is its exclusive concentration on the off-chain social networks of artists, without inclusion of the on-chain transaction graph. We provide a summary of representative on-chain and on-chain/off-chain matching datasets as well as our EX-Graph in Table 1.

Table 1: Summary of datasets featuring on-chain and off-chain matching. EX-Graph stands out due to its open-source nature, heterogeneity, and notably, it has the most extensive collection of on-chain and off-chain matching links compared to any other available datasets.

| Datasets | Open-sourced | Hetero-geneous | # Matching Links | # On-chain Nodes | # On-chain Edges | # Off-chain Nodes | # Off-chain Edges | # Wash-trading Addresses |
|---|---|---|---|---|---|---|---|---|
| D1 (Huang et al., 2022) | × | × | 0 | 1,402,220 | 2,815,028 | 0 | 0 | 0 |
| D2 (Li et al., 2022) | × | × | 0 | 6,844,050 | 208,847,461 | 0 | 0 | 0 |
| D3 (Lin et al., 2022) | ✓ | × | 0 | 40,635 | 1,095,245 | 0 | 0 | 0 |
| D4 (Shamsi et al., 2022) | ✓ | × | 0 | 480,000 | 1,000,000 | 0 | 0 | 6,192 |
| D5 (Béres et al., 2021) | ✓ | × | 890 | 159,399 | 1,155,188 | 890 | 0 | 0 |
| D6 (Vasan et al., 2022) | ✓ | × | 13,487 | 13487 | 0 | 14,706 | 14,066 | 0 |
| **EX-Graph (ours)** | ✓ | ✓ | 30,387 | 2,610,465 | 29,585,858 | 1,103,509 | 3,768,281 | 1,445 |

## 3 PROPOSED DATASET

To enhance the analysis of on-chain data by incorporating off-chain information, we present EX-Graph (short for "Ethereum-X Graph"), a pioneering and large-scale open-source dataset integrating Ethereum transaction data with X following data. This section details our data collection process and graph construction, visually presented in Figure 2.

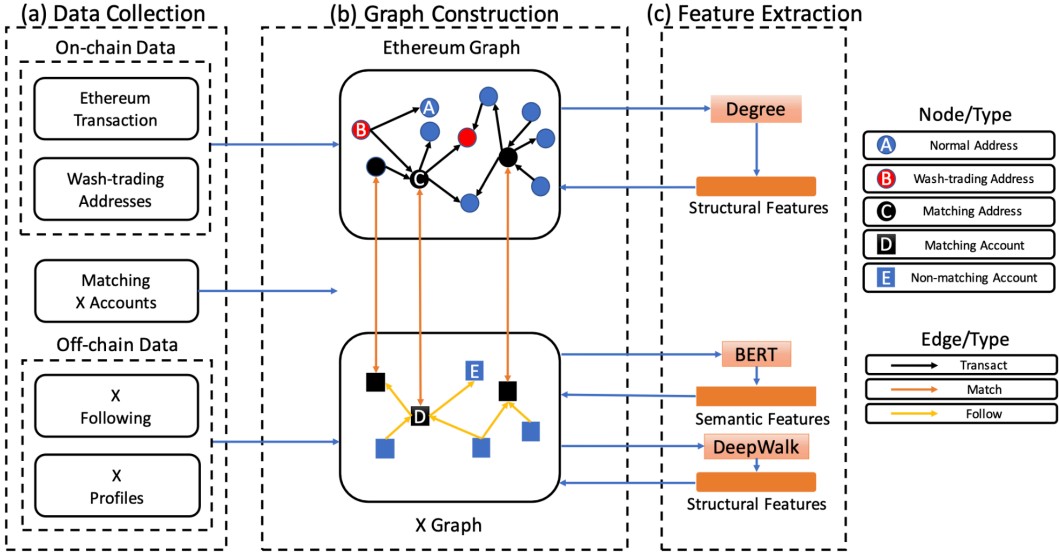

Figure 2: Graph construction from our dataset. This process initiates with the collection of on-chain, matching, and off-chain data, from which we construct on-chain and off-chain graphs. Subsequently, we extract features for Ethereum addresses and X accounts.

## 3.1 DATA COLLECTION

In this section, we outline the process of collecting on-chain, matching, and off-chain data. Throughout this data collection process, we rigorously adhere to the privacy policies of OpenSea and X, ensuring utmost respect and protection for user privacy. For more details regarding our ethical considerations, please refer to Appendix D.

**On-chain Data:** We acquire the required Ethereum transaction records by deploying a local Ethereum Virtual Machine (EVM) client, the runtime environment for executing smart contracts on the Ethereum (Hildenbrandt et al., 2018). This client facilitates the download of an exhaustive dataset encompassing Ethereum transaction records pertaining to NFT from March 1, 2022, to August 31, 2022. This results in 2,610,465 unique Ethereum addresses and 29,585,858 transaction records between them, ensuring broad coverage of on-chain activities. Additionally, we enhance our on-chain data by gathering 1,445 labeled wash-trading Ethereum addresses from Dune [5], enabling us to investigate and detect wash-trading addresses in Ethereum. More details are in Appendix E.

**Matching Data**: We obtain X usernames linked to Ethereum addresses by utilizing the 'User' API endpoint from OpenSea. This data is sourced from OpenSea profile pages, resulting in 30,387 entries that match Ethereum addresses with X accounts. More details are in Appendix F.

**Off-chain Data**: We utilize X's developer API's 'Followers' and 'Friends' endpoints to gather followers and following usernames associated with the selected X accounts. Specifically, 'Followers' are accounts who follow a X account, whereas 'Friends' are accounts that a X account follows. Finally, we collect 1,103,569 X accounts, interconnected by 3,768,281 follower-following relationships. Furthermore, using the 'Users' endpoint of the X developer API, we collect the profiles of all matched X accounts. The total collection accounts to 30,387 X profiles. These X profiles serve as valuable off-chain data, enriching our analysis and complementing the Ethereum transactions. More details are in Appendix G.

## 3.2 GRAPH CONSTRUCTION

All collected data is presented in a heterogeneous graph $G = (V, E, A, R)$. This graph captures the relationships between Ethereum addresses and X accounts, enabling a comprehensive analysis of their activities. The node set $V$ comprises Ethereum addresses and X accounts, i.e., $V = V_{\text{eth}} \cup V_{\text{X}}$.

---

[5]https://dune.com/queries/364051

The edge set $E$ consists of three types of edges: Ethereum transactions, Ethereum-X matches, and X following relationships, i.e., $E = E_{\text{eth}} \cup E_{\text{match}} \cup E_{\text{X}}$. We define two functions $\tau(v) : V \to A$ and $\phi(e) : E \to R$. The function $\tau(v)$ assigns node types, distinguishing Ethereum addresses as "Ethereum", and X accounts as "X". Similarly, the function $\phi(e)$ assigns directed-edge types, categorizing Ethereum transactions as "Transact", Ethereum-X matches as "Match", and X following relationships as "Follow". Importantly, each Ethereum transaction carries temporal information reflecting the timestamp of each transaction.

### 3.3 Feature Extraction

We extract structural and semantic features for nodes. The structural features are sourced from the Ethereum graph, while the semantic features are drawn from the profiles of matched X accounts. Furthermore, we obtain embeddings for all X accounts using the DeepWalk algorithm in the X graph (Yuan et al., 2020). Drawing from the methodology articulated in (Chen et al., 2020), we select a set of eight structural features that holistically represent nodes within the Ethereum transaction graph inspired by these insights. These features are degree, in-degree, out-degree, neighbors, in-neighbors, out-neighbors, the maximum number of transactions with neighbors, and average neighbor degree. An in-depth description of these features are in Appendix C.

We leverage BERT (Devlin et al., 2018) for semantic feature extraction, converting the text profile $s_n$ of each X account $n$'s text profile $s_n$ into a 768-dimensional embedding $h_n$ :

$$h_n = \text{BERT}(s_n), h_n \in \mathbb{R}^{768}, \tag{1}$$

We choose BERT because of its proven effectiveness and widespread use in similar research. More details are in our Appendix C. Then, we enhance this semantic representation by appending the logarithm of the number of followers and followings, resulting in a 770-dimensional embedding $h'_n \in \mathbb{R}^{770}$ :

$$h'_n = h_n \oplus (\log(\# \text{ followers} + 1), \log(\# \text{ followings} + 1)), h'_n \in \mathbb{R}^{770}, \tag{2}$$

This extended feature set is then processed. Each node in the Ethereum transaction and X following graph is equipped with an 8-dimensional structural feature set. Furthermore, every matched X account now possesses a 770-dimensional semantic feature set. To maintain uniformity across diverse feature sets and ensure a balanced model influence, we implement Principal Component Analysis (PCA). This adjustment effectively reduces the dimensionality of the X semantic features to 8, denoted as $h''_n \in \mathbb{R}^8$ :

$$h''_n = \text{PCA}(h'_n), h''_n \in \mathbb{R}^8. \tag{3}$$

## 4 Statistics and Experiments

In this section, we first do an in-depth statistical analysis of our carefully curated EX-Graph dataset in Section 4.1. The focus of this analysis is the structural features mentioned in Section 3.3 of Ethereum addresses that are matched with X accounts versus those that are not, as well as differentiating wash-trading Ethereum addresses from non-wash-trading ones (with the latter being treated as normal Ethereum addresses). Then we conduct comprehensive experiments to address the following three research questions:

***Q1:*** **To what extent does X auxiliary information improve the performance of link prediction in Ethereum transactions?**

***Q2:*** **How significantly does X auxiliary information enhance the detection of wash-trading addresses in Ethereum transactions?**

***Q3:*** **Is it feasible to predict additional matching links based on known matching links?**

### 4.1 EX-GRAPH STATISTICAL ANALYSIS

#### 4.1.1 OVERALL ANALYSIS

Table 2: Summary of EX-Graph. # Wash-trading Addresses listed in the X row means how many wash-trading addresses match to X accounts.

| Graph | # Nodes | # Edges | Avg. Degree | # Matched Accounts | # Wash-trading Addresses |
|---|---|---|---|---|---|
| Ethereum | 2,610,465 | 29,585,858 | 11.33 | 30,667 | 1,445 |
| X | 1,103,509 | 3,768,281 | 3.42 | 30,667 | 3 |

EX-Graph combines the Ethereum and X graphs, as shown in Table 2. Ethereum has a higher average degree compared with X graph. However, the quantity of Ethereum addresses that match with a X account is notably small. The limited wash-trading addresses in the dataset highlight data imbalance, which is often seen in financial data. Only three wash-trading addresses match with X, reflecting the hidden nature of wash-trading activities. Overall, EX-Graph gives a glimpse into Ethereum transactions and wash-trading behaviors, highlighting the difficulty in detecting wash-trading addresses in Ethereum.

#### 4.1.2 ETHEREUM ADDRESSES BREAKDOWN

In terms of the eight structural features as mentioned in Section 3.3, in Table 3 we statistically compare the 30,667 X-matched Ethereum addresses with the rest of the Ethereum addresses. Similarly, for Ethereum graph, we further compare the 1,445 wash-trading Ethereum addresses with the other normal addresses, as shown in Table 4.

Table 3: Comparison between X-matched and non-X-matched Addresses.

| Features | X-matched Addresses | | | Non-X-matched Addresses | | |
|---|---|---|---|---|---|---|
| | Average | Median | Std | Average | Median | Std |
| Degree | 28.26 | 2.00 | 492.39 | 12.98 | 2.00 | 80.75 |
| In-degree | 15.24 | 1.00 | 309.76 | 6.48 | 1.00 | 38.55 |
| Out-degree | 13.03 | 1.00 | 185.41 | 6.50 | 1.00 | 51.65 |
| Neighbors | 26.58 | 2.00 | 402.22 | 12.64 | 2.00 | 73.85 |
| In-neighbors | 15.24 | 1.00 | 309.76 | 6.48 | 1.00 | 38.55 |
| Out-neighbors | 13.03 | 1.00 | 185.41 | 6.50 | 1.00 | 51.65 |
| Max. txns with neighbors | 6.78 | 3.00 | 27.94 | 2.89 | 1.00 | 30.42 |
| Average neighbor degree | 33076.00 | 31105.75 | 27921.99 | 1442.87 | 149.00 | 7170.93 |

Table 4: Comparison between wash-trading and normal addresses on Ethereum graph.

| Features | Wash-Trading Addresses | | | Normal Addresses | | |
|---|---|---|---|---|---|---|
| | Average | Median | Std | Average | Median | Std |
| Degree | 137.61 | 15.00 | 377.70 | 13.02 | 2.00 | 92.74 |
| In-degree | 50.53 | 7.00 | 115.58 | 6.52 | 1.00 | 48.46 |
| Out-degree | 87.09 | 6.00 | 283.29 | 6.50 | 1.00 | 53.83 |
| Neighbors | 134.66 | 14.00 | 371.35 | 12.67 | 2.00 | 82.41 |
| In-neighbors | 87.09 | 6.00 | 283.29 | 6.50 | 1.00 | 53.83 |
| Out-neighbors | 50.53 | 7.00 | 115.58 | 6.52 | 1.00 | 48.46 |
| Max. txns with neighbors | 18.38 | 8.00 | 55.60 | 2.91 | 1.00 | 30.37 |
| Average neighbor degree | 423.95 | 247.84 | 986.01 | 1740.28 | 151.71 | 8220.95 |

From Tables 3 and 4, it is clear that X-matched and wash-trading Ethereum addresses show higher activity levels across the first seven features compared to others. The increased in-degree, out-degree, and maximum transactions with neighbors suggest X-matched addresses are part of an active sub-network within Ethereum. Wash-trading addresses' higher degree, and number of neighbors underscore their active transaction behavior. These addresses exhibit higher median in-degree, out-degree values, and a larger average degree, indicating significant transaction volumes.

The significant average neighbor degree for X-matched addresses primarily reflects frequent transactions with high-degree nodes, such as smart contracts in activities like NFT minting (Wang et al., 2021). Contrastingly, wash-trading addresses exhibit a lower average neighbor degree, likely due to their circular transactions among a limited group of addresses. This behavior restricts interaction with the high-degree nodes of the wider Ethereum network (Victor & Weintraud, 2021; Luo et al., 2023). We provide a more detailed analysis in Appendix H.

In conclusion, we highlight notable differences in transaction activities and network centrality between X-matched and non-X-matched Ethereum addresses, as well as between wash-trading and normal Ethereum addresses. In the following experiments, we explore the correlation between these structural features and Ethereum transaction link prediction, with special focus on the classification of Ethereum addresses, especially wash-trading addresses.

## 4.2 ETHEREUM LINK PREDICTION (*Q1*)

Ethereum link prediction can help monitor transaction trends of Ethereum. However, previous works only research on Ethereum dataset (Lin et al., 2021) (Lin et al., 2022). Therefore, we aim to compare the performance between scenarios: one with the addition of X features and one without. We employ eleven baseline models: DeepWalk (Perozzi et al., 2014), Node2Vec (Grover & Leskovec, 2016), GGNN (Li et al., 2015), GCN (Kipf & Welling, 2016), GAT (Veličković et al., 2017), GraphSage (Hamilton et al., 2017), APPNP (Gasteiger et al., 2018), TAGCN (Du et al., 2017), Cluster-GCN (Chiang et al., 2019), DAGNN (Liu et al., 2020), and GATv2 (Brody et al., 2021).

**Implementation details.** Firstly, We divide all Ethereum graph edges into training, validation, and test sets, following a 70/10/20 ratio based on edge timestamps. We then remove the validation and test edges and choose the largest connected subgraph from the remaining edges to constitute our training graph. This process ensures that we only use old links to predict new ones. Next, we focus on all edges within our training graph that are connected to at least one X-matched Ethereum address to serve as positive training edges. For balance, we generate an equal number of negative training edges by selecting a non-existent link from each X-matched Ethereum address within the positive training edges. For the validation and test sets, we choose positive edges from the previously partitioned validation and test sets and ensure that both nodes in a validation or test edge are also present in the training graph.

For every positive edge within the validation and test sets, we randomly choose a non-existent link stemming from the same X-associated Ethereum address. We provide details of this setup in Appendix I. With this framework, we execute two experiments: one that incorporates X features and one that does not. Our goal is to compare the results of these two scenarios to determine the influence of X data on the analysis of Ethereum activities.

**Evaluation metrics.** We use the following four widely used metrics (Chen et al., 2020), to evaluate the performance of experiments in a comprehensive way: **(1) AUC-ROC**, **(2) Precision**, **(3) Recall**, and **(4) F1-score.** Note that, for all the experiments we repeat 5 times and report the average performance with standard deviation.

Table 5: Results for Ethereum link prediction, with ↑, -, and ↓ indicating performance increase, maintaining and decrease, respectively.

| Method | Without X Features | | | | With X Features | | | |
|---|---|---|---|---|---|---|---|---|
| | AUC-ROC | Precision | Recall | F1 | AUC-ROC | Precision | Recall | F1 |
| DeepWalk | 0.72±0.00 | 0.81±0.00 | 0.57±0.00 | 0.67±0.00 | 0.74±0.00 (↑) | 0.83±0.00 (↑) | 0.60±0.00 (↑) | 0.70±0.00 (↑) |
| Node2Vec | 0.72±0.00 | 0.81±0.00 | 0.57±0.00 | 0.67±0.00 | 0.74±0.00 (↑) | 0.83±0.00 (↑) | 0.60±0.00 (↑) | 0.70±0.00 (↑) |
| GCN | 0.76±0.01 | 0.70±0.01 | 0.67±0.02 | 0.69±0.01 | 0.81±0.00 (↑) | 0.81±0.01 (↑) | 0.77±0.02 (↑) | 0.75±0.01 (↑) |
| GAT | 0.77±0.01 | 0.73±0.02 | 0.67±0.02 | 0.70±0.01 | 0.81±0.02 (↑) | 0.75±0.01 (↑) | 0.76±0.01 (↑) | 0.75±0.01 (↑) |
| GATv2 | 0.77±0.01 | 0.72±0.01 | 0.74±0.02 | 0.73±0.01 | 0.79±0.00 (↑) | 0.73±0.01 (↑) | 0.75±0.01 (↑) | 0.74±0.00 (↑) |
| GraphSage | 0.84±0.00 | 0.77±0.01 | 0.77±0.01 | 0.78±0.00 | 0.86±0.00 (↑) | 0.81±0.01 (↑) | 0.79±0.01 (↑) | 0.80±0.00 (↑) |
| TAGCN | 0.81±0.01 | 0.82±0.01 | 0.64±0.02 | 0.72±0.01 | 0.88±0.00 (↑) | 0.85±0.01 (↑) | 0.73±0.01 (↑) | 0.78±0.00 (↑) |
| Cluster-GCN | 0.82±0.01 | 0.78±0.01 | 0.70±0.01 | 0.74±0.01 | 0.86±0.00 (↑) | 0.81±0.00 (↑) | 0.75±0.01 (↑) | 0.78±0.00 (↑) |
| DAGNN | 0.79±0.00 | 0.77±0.01 | 0.67±0.01 | 0.72±0.00 | 0.87±0.00 (↑) | 0.82±0.00 (↑) | 0.77±0.01 (↑) | 0.79±0.00 (↑) |
| APPNP | 0.82±0.01 | 0.80±0.03 | 0.68±0.02 | 0.74±0.01 | 0.89±0.02 (↑) | 0.83±0.00 (↑) | 0.78±0.00 (↑) | 0.81±0.00 (↑) |
| GGNN | 0.82±0.01 | 0.81±0.01 | 0.68±0.01 | 0.74±0.00 | 0.84±0.00 (↑) | 0.79±0.01 (↑) | 0.73±0.02 (↑) | 0.76±0.01 (↑) |

**Discussion.** Results shown in Table 5 demonstrate the superiority of deep learning models that leverage both graph structures and X features compared to those that rely solely on graph structures. Notably, incorporating features from matched X accounts significantly improves the performance of all models, as evidenced by enhanced metrics. For instance, APPNP exhibits a substantial 7% increase in AUC-ROC when integrating X features, while DAGNN shows an 8% increase in AUC-ROC and a remarkable 10% surge in recall. This improvement is attributed to the fact that by incorporating X data, we gain valuable insights into the behavior and characteristics of these addresses that are not available from Ethereum transaction data alone.

### 4.3 WASH-TRADING ADDRESSES DETECTION (*Q2*)

Wash trading is a common fraudulent practice on Ethereum. Its usual objectives are manipulating a cryptocurrency's price and facilitating money laundering (Cao et al., 2015). Detecting wash-trading addresses can help protect the honest users on Ethereum. To assess the effectiveness of including X information in detecting these addresses, we employ the same eleven baseline models in Section I.4.

**Implementation details.** First, we split the whole Ethereum graph into training, validation, and testing graphs following a 70/10/20 ratio based on edge timestamps. This ensures temporal consistency and upholds the transaction sequence, as node features stem from their edges and neighbors. Then we keep the largest connected subgraph of the training graph for training baseline models.

Given the significant imbalance between the wash-trading Ethereum addresses and the normal Ethereum addresses - approximately 0.05% and 99.95% respectively - we adopt a balanced sampling strategy. Specifically, all wash-trading addresses that appear in the training, validation, and testing graphs are considered as positive samples. We randomly choose an identical number of normal addresses from each of the train/validation/test graphs as negative samples. This strategy ensures a balanced positive and negative sample distribution. Under this setup, we conduct two experiments for each model - one that incorporates X features and one that does not. The objective is to compare the performance between these two situations to see how much X can affect the classification of wash-trading Ethereum addresses. More implementation details are in Appendix J.

**Evaluation metrics.** We employ the following four widely used metrics: **(1) AUC-ROC**, **(2) Precision**, **(3) Recall**, and **(4) F1-score**. Among these metrics, we place a higher emphasis on **recall**, as it depicts the percentage of detected wash-trading addresses out of all wash-trading addresses (Chen et al., 2020). Prioritizing recall underscores a preference for navigating false risk warnings over experiencing deception. For each experiment, we repeat it 5 times and report the average performance along with the standard deviation.

Table 6: Results for wash-trading addresses detection.

| Method | Without X Features | | | | With X Features | | | |
|---|---|---|---|---|---|---|---|---|
| | AUC-ROC | Precision | Recall | F1 | AUC-ROC | Precision | Recall | F1 |
| DeepWalk | 0.54±0.00 | 0.54±0.00 | 0.59±0.00 | 0.56±0.00 | 0.55±0.00 (↑) | 0.55±0.00 (↑) | 0.58±0.00 (↓) | 0.57±0.00 (↑) |
| Node2Vec | 0.54±0.00 | 0.54±0.00 | 0.59±0.00 | 0.56±0.00 | 0.55±0.00 (↑) | 0.55±0.00 (↑) | 0.58±0.00 (↓) | 0.57±0.00 (↑) |
| GCN | 0.79±0.01 | 0.81±0.02 | 0.56±0.01 | 0.66±0.01 | 0.79±0.01 (-) | 0.82±0.01 (↑) | 0.56±0.01 (-) | 0.66±0.01 (-) |
| GAT | 0.80±0.01 | 0.80±0.01 | 0.59±0.00 | 0.68±0.00 | 0.80±0.00 (↑) | 0.74±0.01 (↓) | 0.72±0.00 (↑) | 0.73±0.00 (↑) |
| GATv2 | 0.80±0.00 | 0.76±0.01 | 0.68±0.01 | 0.72±0.00 | 0.81±0.01 (↑) | 0.74±0.02 (↓) | 0.74±0.02 (↑) | 0.74±0.01 (↑) |
| GraphSage | 0.78±0.01 | 0.81±0.01 | 0.52±0.01 | 0.63±0.01 | 0.80±0.01 (↑) | 0.75±0.01 (↓) | 0.70±0.02 (↑) | 0.72±0.01 (↑) |
| TAGCN | 0.80±0.01 | 0.76±0.01 | 0.67±0.01 | 0.71±0.01 | 0.80±0.01 (-) | 0.74±0.01 (↓) | 0.73±0.01 (↑) | 0.73±0.01 (↑) |
| Cluster-GCN | 0.80±0.01 | 0.74±0.01 | 0.69±0.01 | 0.71±0.01 | 0.81±0.00 (↑) | 0.75±0.02 (↑) | 0.72±0.01 (↑) | 0.74±0.00 (↑) |
| DAGNN | 0.80±0.00 | 0.74±0.01 | 0.71±0.00 | 0.72±0.00 | 0.81±0.00 (↑) | 0.75±0.01 (↑) | 0.73±0.02 (↑) | 0.74±0.01 (↑) |
| APPNP | 0.80±0.01 | 0.74±0.01 | 0.70±0.01 | 0.72±0.00 | 0.81±0.00 (↑) | 0.75±0.01 (↑) | 0.73±0.01 (↑) | 0.74±0.00 (↑) |
| GGNN | 0.79±0.01 | 0.75±0.00 | 0.69±0.02 | 0.72±0.01 | 0.80±0.01 (↑) | 0.75±0.01 (↓) | 0.72±0.01 (↑) | 0.73±0.01 (↑) |

**Discussion.** Table 6 reveals that models like DeepWalk and Node2Vec underperform as they do not utilize node features. The relatively low recall in GNN models illustrates the challenge in detecting wash-trading addresses from the heavily imbalanced data as normal addresses significantly outnumber fraudulent ones. The integration of X and structural features notably enhances AUC-ROC, recall, and F1 scores across most models, particularly boosting the recall score by 18% in GraphSage and 13% in GAT. Despite observed trade-offs between precision and recall in some results, our method marks substantial progress in detecting these fraudulent activities, showcasing the benefits of integrating both Ethereum and X data in identifying wash-trading activities in Ethereum.

## 4.4 MATCHING LINK PREDICTION (*Q3*)

In this section, we aim to predict matching links between X accounts and Ethereum addresses to further enrich our dataset. We utilize thirteen baseline models for this task. Among these, DeepWalk (Perozzi et al., 2014), Node2Vec (Grover & Leskovec, 2016), and HIN2Vec (Fu et al., 2017) operate without node features, whereas the others incorporate them. Specifically, GGNN (Li et al., 2015), GCN (Kipf & Welling, 2016), GAT (Veličković et al., 2017), GraphSage (Hamilton et al., 2017), APPNP (Gasteiger et al., 2018), TAGCN (Du et al., 2017), Cluster-GCN (Chiang et al., 2019), DAGNN (Liu et al., 2020), and GATv2 (Brody et al., 2021) are designed for homogeneous graph embeddings. In contrast, HIN2Vec (Fu et al., 2017) and R-GCN (Schlichtkrull et al., 2018) are tailored for heterogeneous graph embeddings.

**Implementation details.** In this study, we focus on the largest connected subgraph extracted from the whole EX-Graph we have created. As our matching edges do not incorporate temporal data, we randomly partition all matching edges into training, validation, and testing sets in a 70/10/20 split respectively. We treat all matching edges as positive samples. For the negative samples, we opt for a random selection approach where we pick non-existing connections between Ethereum addresses and all their matched X accounts. This method ensures we maintain a balanced 1-to-1 ratio between positive and negative edges. Further implementation details can be found in Appendix K.

**Evaluation metrics.** We use the following four widely-used metrics to evaluate the performance of experiments in a comprehensive way: **(1) AUC-ROC**, **(2) Precision**, **(3) F1-score**, and **(4) Accuracy**. For each experiment, we repeat 5 times and report the average performance with standard deviation.

Table 7: Results for matching link prediction (the best results are **bolded**).

| Method | AUC-ROC | Precision | F1 | Accuracy |
|---|---|---|---|---|
| DeepWalk | 0.42±0.00 | 0.31±0.01 | 0.18±0.01 | 0.42±0.00 |
| Node2Vec | 0.42±0.00 | 0.32±0.01 | 0.18±0.01 | 0.42±0.00 |
| GCN | 0.71±0.01 | 0.64±0.01 | 0.65±0.03 | 0.71±0.01 |
| GAT | 0.60±0.00 | 0.52±0.03 | 0.61±0.01 | 0.56±0.01 |
| GATv2 | 0.48±0.01 | 0.50±0.02 | 0.44±0.00 | 0.50±0.01 |
| GraphSage | 0.67±0.02 | 0.63±0.02 | 0.71±0.02 | 0.67±0.01 |
| TAGCN | 0.66±0.03 | 0.63±0.02 | 0.67±0.01 | 0.65±0.01 |
| Cluster-GCN | 0.67±0.01 | 0.65±0.01 | 0.66±0.01 | 0.65±0.01 |
| DAGNN | **0.74±0.00** | 0.66±0.00 | **0.79±0.00** | **0.74±0.00** |
| APPNP | 0.73±0.01 | **0.69±0.00** | 0.72±0.00 | 0.70±0.00 |
| GGNN | 0.71±0.01 | 0.65±0.00 | 0.76±0.01 | 0.71±0.01 |
| R-GCN | 0.66±0.00 | 0.66±0.01 | 0.65±0.00 | 0.66±0.00 |
| HIN2Vec | 0.49±0.00 | 0.49±0.01 | 0.36±0.01 | 0.49±0.00 |

**Discussion.** As illustrated in Table 7, DeepWalk and Node2Vec perform poorly due to their limited model complexity and grasp of node features during training. HIN2Vec, which also doesn't leverage node features, performs suboptimally compared to the other GNN models. DAGNN excels due to its ability to capture large-scale graph information effectively (Liu et al., 2020). Other GNN models yield comparable results across all metrics, suggesting both homogeneous and heterogeneous models are similarly effective for this task. Furthermore, the similar results highlight the considerable potential of GNNs to accurately forecast additional matching links, enhancing the richness of our EX-Graph.

## 5 CONCLUSION

We present EX-Graph, a pioneering and extensive dataset that bridges Ethereum and X. Experiments on EX-Graph highlight integrating additional X data significantly boosts the effectiveness of various tasks on Ethereum. Our research facilitates the understanding of the interplay between on-chain and off-chain worlds and paves the way for novel methods in probing on-chain activities using off-chain supplementary data. However, we recognize that there are still areas to work on, such as collecting more matching links between on-chain and off-chain graphs, and wash-trading Ethereum addresses to counter the data imbalance. Our next steps involve incorporating these enhancements and constantly updating our graph with more on-chain and off-chain nodes and edges.

## 6 ETHICS STATEMENT

Our research adheres rigorously to ethical standards in data collection from both X and OpenSea. We obtained X interaction data in compliance with its guidelines, and associated X accounts with Ethereum addresses using exclusively public information. To ensure privacy, all handles were anonymized, and our data sharing abided by X's academic redistribution policies. Through these meticulous measures, we achieved a responsible balance between insights and privacy across blockchain and social media domains. Our research, focused on public and pre-anonymized data, is under exemption review by the Institutional Review Board (IRB), supervised by the faculty. We remain committed to upholding ethical and privacy guidelines, ensuring our methodologies respect the protections inherent to human subjects.

## 7 REPRODUCIBILITY STATEMENT

We have publicly released our dataset and codebase at `https://github.com/Persdre/EX-Graph`, which includes the datasets, models, and settings necessary to reproduce our results.

## 8 ACKNOWLEDGEMENT

This research is supported by the National Research Foundation, Singapore under its AI Singapore Programme (AISG Award No: AISG2-TC-2021-002).

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

APPENDIX

## A  EXPERIMENTAL ENVIRONMENT

All models in our experiments were implemented using Pytorch 2.0.0 in Python 3.9.16, and run on a robust Linux workstation. This system is equipped with two Intel(R) Xeon(R) Gold 6226R CPUs, each operating at a base frequency of 2.90 GHz and a max turbo frequency of 3.90 GHz. With 16 cores each, capable of supporting 32 threads, these CPUs offer a total of 64 logical CPUs for efficient multitasking and parallel computing. The workstation is further complemented by a potent GPU setup, comprising eight NVIDIA GeForce RTX 3090 GPUs, each providing 24.576 GB of memory. The operation of these GPUs is managed by the NVIDIA-SMI 525.60.13 driver and CUDA 12.0, ensuring optimal computational performance for our tasks.

To implement and train our graph-based models, we utilized the version 1.1.0 of the Deep Graph Library (DGL) [6], which is a popular open-source library designed to simplify the development of deep learning models for graph-structured data. By leveraging DGL's optimized performance and flexible APIs, we were able to easily construct, manipulate, and train graph neural networks (GNNs) on large-scale graphs.

## B  SUPPLEMENTAL RELATED WORK

**Graph representation learning.** Graphs are powerful tools for representing complex data and relationships, as they can effectively encapsulate vast amounts of information (Cao et al., 2016). Among various methods for graph analytics, graph representation learning stands out for its ability to convert high-dimensional graph data into a lower-dimensional space while retaining essential information (Cai et al., 2018). Two major categories of graph representation learning techniques emerge, including those based on random walks and deep learning (Goyal & Ferrara, 2018). Random walk methods such as DeepWalk (Perozzi et al., 2014) and Node2Vec (Grover & Leskovec, 2016) maintain higher-order proximity between nodes. They achieve this by maximizing the likelihood of consecutive nodes appearing in random walks of fixed lengths, thereby approximating properties in large graphs (Goyal & Ferrara, 2018). On the other hand, deep learning-based graph representations, such as Graph Convolutional Network (GCN) (Kipf & Welling, 2016), Graph Attention Network (GAT) (Veličković et al., 2017), and GraphSAGE (Hamilton et al., 2017), utilize iterative convolution operators to aggregate neighbor embeddings. This approach enables scalable learning of node representations, capturing the global characteristics of the neighborhood (Goyal & Ferrara, 2018). For homogeneous graphs, popular embedding methods include DeepWalk (Perozzi et al., 2014), Node2Vec (Grover & Leskovec, 2016), GCN (Kipf & Welling, 2016), GAT (Veličković et al., 2017), and GraphSage (Hamilton et al., 2017). For heterogeneous graphs, methods such as HIN2Vec (Fu et al., 2017), Relational Graph Convolutional Network (R-GCN) (Schlichtkrull et al., 2018), Heterogeneous Graph Attention Network (HAN) (Wang et al., 2019), Heterogeneous Graph Transformer (HGT) (Hu et al., 2020), and Metapath Aggregated Graph Neural Network (MAGNN) (Fu et al., 2020) have been proposed and utilized. These embedding methods have been widely used in numerous downstream tasks, such as link prediction (Al Hasan et al., 2006) and node classification (Rong et al., 2019). Furthermore, to meet the privacy protection requirements in graph-based data such as X, many graph representation learning approaches utilize federated learning methods (Tang et al., 2022a;b).

## C  FEATURE EXTRACTION

In this section, we elaborate on the feature extraction process, detailing the nature and importance of each feature extracted for our study.

### C.1  STRUCTURAL FEATURE EXTRACTION

Structural features consist of two parts: DeepWalk embeddings, and eight statistical features.

---

[6] https://github.com/dmlc/dgl.

**DeepWalk.** In our study, we deploy the DeepWalk model to generate embeddings that capture the structural features of the X network. Each X account is represented within a 128-dimensional embedding space, striking a balance between detail and computational efficiency. The model navigates through the network with a walk length of 40 and a context size of 5, allowing for an extensive exploration of the graph and a focus on local node interactions. We have set the number of walks per node to 10, ensuring diverse and robust random walks for each node. Additionally, for an effective learning process, the model employs a strategy of using a single negative sample against each positive instance. This configuration of the DeepWalk model is integral to our approach in accurately representing the complex relationships within the X network.

**Degree.** This is a measure of the node's overall connectivity, representing the total number of transactions either sent or received by the node. It provides insights into the node's activity level within the Ethereum network.

**In-degree.** This feature reflects the number of incoming transactions to the node from other nodes. A high in-degree indicates a node of significant interest or utility within the network.

**Out-degree.** This counts the number of transactions that the node initiates, i.e., sends out to other nodes. A high out-degree could suggest an active node that frequently interacts with other nodes.

**Neighbors.** This feature records the total number of unique nodes with which the node has a connection, be it via incoming or outgoing transactions. It provides a sense of the node's direct reach within the network.

**In-neighbors.** This is the count of unique nodes that have transactions with the current node, illustrating the diversity of nodes sending transactions to the node.

**Out-neighbors.** This is the number of unique nodes to which the current node sends transactions, illustrating the diversity of nodes with which the node interacts.

**Maximum number of transactions with neighbors.** This feature reflects the highest transaction frequency the node has with any of its neighbors, providing insight into the node's strongest connection in terms of transaction count.

**Average neighbor degree.** This refers to the mean degree of all the node's neighbors. It gives a sense of the general connectivity and activity level of the nodes directly linked to the node in the graph.

## C.2 X SEMANTIC FEATURE EXTRACTION

We leverage BERT (Devlin et al., 2018) for semantic feature extraction, transforming each X profile into a 768-dimensional embedding. We choose BERT for the following two reasons.

**Proven efficacy in text representation:** BERT has been widely recognized for its ability to generate semantically rich representations of textual data. This is primarily due to its bidirectional nature, which enables it to consider both previous and upcoming words in a given text. The embeddings it produces are often dense with information, making them well-suited for tasks like ours that require nuanced understanding of content.

**Prevalence in similar research:** Several peer-reviewed studies have leveraged BERT for feature extraction from textual data. For instance, Qiu et al. (2020) have harnessed BERT's capabilities for extracting contextualized features in various domains, showcasing its versatility and applicability. Sawhney et al. (2022) have used BERT to extract text embeddings from NFT-related tweets.

## D DATA ETHICS

### D.1 OPENSEA DATA COLLECTION

To obtain X handles associated with Ethereum addresses, we utilize OpenSea's public API endpoint "User". Our data collection methods strictly adhere to OpenSea's Terms of Service [7] and privacy policy [8].

---

[7]https://opensea.io/tos.
[8]https://opensea.io/privacy.

According to these terms, any information or data that users add to their accounts, including usernames, preferred items, watchlisted collections, and other provided details, may be collected. Additionally, **social media accounts** such as X accounts linked to the user can also be collected. It is important to note that public content may remain accessible on the internet following its deletion from an OpenSea account.

Therefore, it is both possible and ethical for us to gather X handles connected to Ethereum addresses from OpenSea user profiles. However, before releasing any data collected from OpenSea, we take steps to ensure that all user data is anonymized to protect users' privacy.

By adhering to these policies and practices, we can collect valuable data for our analysis while respecting the privacy of individuals.

## D.2 X DATA COLLECTION

Our collection of X interaction data is carried out ethically and in compliance with X's policies. We gather information on following relationships between handles formatted as one handle following another, as well as user profiles using the "Friends" and "User" API endpoints provided by X developers.

To ensure compliance with X's off-X matching policy [9], we are very careful when matching X accounts with Ethereum addresses. This policy restricts us to using only publicly available data for matching attempts, which includes profile information such as the account bio and publicly disclosed location, as well as the display name and *@handle*. It is important to note that the display name can be changed freely by X users at any time, while the *@handle* is unique to each X user, and is commonly referred to as the user name.

By adhering to this policy, we ensure that our approach to matching X accounts with Ethereum addresses is both ethical and effective. While we only have access to limited public information, we can still achieve a high degree of accuracy in our matching attempts. However, it is important to note that due to the limitations of this approach, there may be cases where we are unable to match a X account with an associated Ethereum address.

To ensure that our data is open-source, we closely follow X's content redistribution policies designed for academic researchers. These policies permit academic researchers associated with an academic institution and engaged in non-commercial research to distribute an unlimited number of Tweet IDs and/or User IDs. This enables us to share an unlimited amount of User IDs for purposes such as facilitating peer review or validating research. We have made the X following graph open-source, anonymizing all handles by substituting them with their numeric IDs in the graph. Additionally, we provide open-source BERT-handled vectors consisting solely of numerical data for X profiles, ensuring that no word information from the original profiles is disclosed.

## E ON-CHAIN DATA COLLECTION

Though many previous works utilized third-party API such as Etherscan blockchain explorer API to collect Ethereum transaction records (Lin et al., 2020; 2021; 2022; Wang et al., 2022), we choose to instantiate a local Ethereum Virtual Machine (EVM) to acquire complete Ethereum transaction records. This approach ensures we obtain a thorough dataset encompassing all on-chain activities without depending on third-party platforms.

The first step involves setting up an Ethereum node on a local machine. We use Geth [10] which synchronizes with the Ethereum blockchain network. We choose Geth's full node type that provides a more comprehensive dataset but requires substantial storage space as it downloads the entire blockchain.

After successfully establishing the local node, we start the process of extracting data. Ethereum transactions are packaged into blocks that are appended to the Ethereum blockchain. These blocks contain various pieces of data, including the sender's address, receiver's address, value transferred,

---

[9] `https://developer.X.com/en/developer-terms/policy`.
[10] `https://geth.ethereum.org`.

gas used, timestamp, and more. Using the local geth node's API, we parse these blocks and extract relevant transaction data. One example is below:

For example, when we parse a block, the data structure may resemble the following:

```
{
    "number": "4370000",
    "hash": "0x5eaa2...",
    "parentHash": "0x4eaa3...",
    "nonce": "0x04668f72247a130c",
    "sha3Uncles": "0x1dcc...",
    "logsBloom": "0x...",
    "transactionsRoot": "0x9a4f1f...",
    "stateRoot": "0x5f5f8f...",
    "miner": "0x2a6...",
    "difficulty": "2262461398322124",
    "totalDifficulty": "1577795892494985633613",
    "size": "13606",
    "extraData": "0x74657374",
    "gasLimit": "6712388",
    "gasUsed": "21000",
    "timestamp": "1502133250",
    "transactions": [ ... ],
    "uncles": [ ... ]
}
```

The "transactions" field in a block on the Ethereum blockchain contains an array of transaction objects, each representing a specific transaction that occurred within that block. These transaction objects typically include critical information such as the sender's address ("from"), the recipient's address ("to"), the value transferred ("value"), and the gas used ("gas"), among other details. Additionally, the block's "timestamp" field provides us with the time when the block was mined and the "block_number" is the unique identifier assigned to that specific block by the network.

By parsing this data, we can construct a comprehensive record of all transactions that have occurred on the Ethereum blockchain, enabling us to analyze patterns in on-chain activities and link them with off-chain data to gain deeper insights into the behavior of Ethereum users.

To obtain a comprehensive dataset for our study, we continuously synchronize the local EVM with the Ethereum blockchain network, ensuring that we collect up-to-date data. This method facilitates the tracking of evolving patterns in on-chain activities and helps us maintain an expansive and current dataset.

For this study, we filter the Ethereum transaction data from March 2022 to August 2022 to focus on a specific timeframe. We then narrow down the dataset by filtering out all NFT transaction records using the transaction hash value as a filter criterion [11]. By doing so, we can efficiently extract all relevant transaction records related to NFT trading during this period for further analysis.

## F    MATCHING DATA COLLECTION

To collect matching data between Ethereum addresses and OpenSea user profiles, we utilize OpenSea's 'User' endpoint API. This API takes an Ethereum address as input and returns the associated OpenSea user profile information, if available. By using this API, we can efficiently match Ethereum addresses with OpenSea user profiles and extract relevant data for our analysis. It is important to note that this process relies on the accuracy and completeness of the data provided by OpenSea, which may be subject to limitations or errors.

For example, when we use the Ethereum address '0xBd0b37e45f940c08175aB994bb8D6F7e473445FA' to query the API, the call looks like this:

"https://api.opensea.io/api/v1/user/0xBd0b37e45f940c08175aB994bb8D6F7e473445FA".

---

[11]https://ethereum.org/en/developers/docs/transactions/.

The API then returns a response that contains the corresponding OpenSea user profile information, including any connected X accounts, if applicable. The response is as follows here:

```
{
    "username": "xx...",
    "account": {
        "user": {
            "username": "xx..."
        },
        "profile_img_url": "https://i.seadn.io/gae...",
        "address": "0xbd0b37e45f940c08175ab994bb8d6f7e473445fa",
        "config": "",
        "currencies": {},
        "profile_image_url": "https://i.seadn.io/gae...",
        "website_url": "",
        "X_username": "xx..."
    }
}
```

In this study, we take great care to anonymize OpenSea user's connected X account information to protect their privacy. It is important to note that the linkage between a X account and an OpenSea user requires verification from OpenSea, which provides us with ground-truth information for our matching attempts.

By examining the response data, specifically the 'X_username' field, we are able to extract a matched X account associated with a given Ethereum address. This process enables us to establish reliable mappings between X accounts and Ethereum addresses while preserving user anonymity.

Overall, our approach respects user privacy while providing us with accurate data for our analysis of the relationships between X users and the NFT market on the Ethereum blockchain.

## G    OFF-CHAIN DATA COLLECTION

During the early phases of our investigation, we embarked on a comprehensive assessment of various potential external data sources, including X, Instagram, Facebook, Discord, and Reddit. Our selection criteria for these platforms hinged on their user activity levels and their apparent relevance to Ethereum interactions. Of these platforms, X stood out due to its heightened engagement with discussions surrounding Ethereum and NFTs. Additionally, X boasts a developer-centric API that simplifies the extraction of user relationships, particularly "following" connections. Beyond mere connections, X's extensive user profile data serves as a gold mine, providing a broad and deep spectrum of information pivotal to our analysis. In juxtaposition, platforms such as Instagram, Facebook, Discord, and Reddit either do not offer as detailed APIs or are fundamentally more reserved in their user interactions, making data procurement and analysis more cumbersome.

Another compelling factor underpinning our selection was the availability of an API from OpenSea, the premier NFT marketplace. This API provision enabled us to establish direct and verified linkages between X accounts and their respective OpenSea profiles. Such integration not only bridged the gap between on-chain and off-chain data but also reinforced our conviction in centering our efforts on X.

To accumulate X's follower-following relationships, we leverage X's Developer API in this study [12]. We begin by using the 'User' endpoint to fetch user profile details using the account user name. Then, for each retrieved user, we obtain their followers and the accounts they are following through the 'Followers' and 'Friends' endpoints, respectively.

The 'Followers' endpoint returns a list of user objects for every user following the specified user, while the 'Friends' endpoint returns a list of user objects followed by the specified user. By extracting this information, we construct a graph of the follower-following relationships.

---

[12]https://developer.X.com/en/docs/X-api.

It is important to note that we strictly adhere to X's API usage policies to ensure the privacy and security of user data. To maintain anonymity when open-sourcing the collected data, we anonymize X user names with their numerical IDs.

This method offers several advantages as it provides a wealth of additional information, including publicly shared profile information, tweets, retweets, and likes, which could provide valuable features for our future analysis. Despite the limitations imposed by the API rate limits, we were able to collect substantial off-chain data that forms a critical part of our dataset.

By incorporating this off-chain data collection method, we can gain a more comprehensive understanding of Ethereum address activities by adding an essential layer of social network context to the on-chain transactional data.

## H    STATISTICAL ANALYSIS

In this section, we conduct a detailed analysis of the statistical results for F8 (Average neighbor degree) measurement of both X-matched Ethereum addresses and wash-trading Ethereum addresses to gain deeper insights into their behavior on the Ethereum network.

### H.1    NFT MINTING PROCESS

After observing an unusually high average neighbor degree among Ethereum addresses that match with X accounts, we conducted a thorough investigation to understand the root causes. Our findings indicate that this phenomenon is largely driven by the process of minting Non-Fungible Tokens (NFTs) on the Ethereum blockchain (Ghelani, 2022).

To mint NFTs on Ethereum, a smart contract is typically deployed to manage the minting process, and each user who mints an NFT interacts with this contract, resulting in a transaction being added to the blockchain. The unique nature of NFT minting means that a large number of Ethereum addresses (users) interact with a relatively small number of smart contracts. As a result, these smart contracts, which are frequently used by Ethereum addresses matched to X accounts, have a significantly higher neighbor degree, causing the overall average neighbor degree for this subset of Ethereum addresses to rise.

Additionally, many of these X-matched Ethereum addresses belong to artists or NFT creators who are highly active in the community, including minting NFTs and interacting with smart contracts. This increased activity leads to a higher frequency of transactions and drives up their neighbor degree even further.

Our discovery highlights the critical role of integrating off-chain information to gain a comprehensive understanding of on-chain activities. It underscores how social media influence and engagement can have a significant impact on on-chain behavior, further emphasizing the value of our integrated dataset in exploring such dynamics.

### H.2    WASH TRADING PATTERNS

When comparing X-matched Ethereum addresses with wash-trading Ethereum addresses, we observe a distinctly lower average neighbor degree (F8 feature) for the latter. This is because that wash-trading Ethereum addresses, by nature, engage in a more closed network of transactions (von Wachter et al., 2022).

In a typical wash trading scheme, a single entity or a coordinated group of entities engage in deceptive transactions where no actual change of ownership occurs. We list some typical wash trading patterns in Figure III. We explain the four patterns presented in this figure as follows:

- In pattern (a), two Ethereum addresses engage in a circular flow of funds, where one address sends Ether to the other address, which then sends it back to the first address. This cycle of transfers creates the illusion of high trading volume while no actual trades are taking place.
- In pattern (b), three Ethereum addresses form a triangle of transfer activities, where each address sends Ether to another address in the cycle. Again, this creates the appearance of high trading volume without any actual trades.

- Pattern (c) involves three Ethereum addresses participating in a cycle of transfers, where one address sends Ether to another address, which then sends it to a third address, and finally, the third address sends it back to its sender. This type of pattern is similar to pattern (b). The purpose of this pattern is to create the illusion of high trading activity without any actual trades taking place.

- Finally, pattern (d) is a variant of pattern (b), where four Ethereum addresses participate in a circular flow of funds. Again, this creates the illusion of high trading activity without any real trading happening.

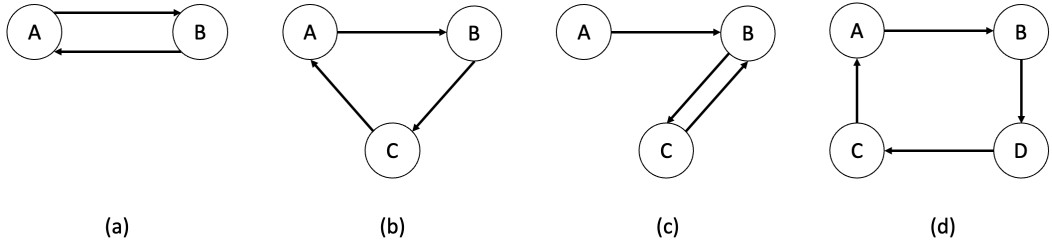

(a)        (b)        (c)        (d)

Figure III: Four typical wash trading patterns. Nodes A, B, C and D represent Ethereum addresses involved in a cycle of transfer activities.

We can see that the transaction patterns of wash-trading addresses result in a lower average neighbor degree as these addresses tend to transact exclusively amongst themselves, limiting their interactions with a broader network of nodes and smart contracts discussed in Section H.1. As a result, they form a smaller, tightly-knit network of interconnected addresses.

Understanding this pattern is crucial for detecting and preventing wash trading activities on the Ethereum network. Incorporating these insights into our dataset strengthens our understanding of Ethereum network dynamics and helps to develop more efficient detection models to combat illicit activities such as wash trading.

# I  ETHEREUM LINK PREDICTION

## I.1  FEATURE ENGINEERING

Our methodology for Ethereum link prediction involves utilizing eight structural features derived from Ethereum addresses, as outlined in Section 4.1 of the main paper. These features serve as the primary input for our modeling approach and help us to accurately predict links between Ethereum wallets.

For DeepWalk and Node2Vec models, we generate features through a combination of random walks on the transaction graph and semantic features associated with matched X accounts. This comparative study allows us to assess the degree of enhancement in the models' predictive power when off-chain social information is incorporated.

For the rest Graph Neural Network (GNN) models, a similar approach is followed. We gauge the effectiveness of the models when trained on structural features alone and contrast it with their performance when trained on both structural and corresponding X account semantic features. This comparison serves to underline the value added by off-chain data in on-chain prediction tasks.

The training of each model yields node embeddings, which essentially represent a compressed form of the nodes. Each model's performance is averaged across five separate experiments, with the standard deviations also being reported to capture the variability in outcomes.

## I.2  MODEL DESCRIPTIONS

In this section, we introduce the models that we utilized as baselines for the Ethereum link prediction task.

- DeepWalk (Perozzi et al., 2014): DeepWalk is a popular graph embedding algorithm that uses random walks to generate node embeddings in a homogeneous network. In DeepWalk, each node is treated as a word in a corpus, and a sequence of nodes obtained through random walks is used to learn vector representations of the nodes.

- GGNN (Li et al., 2015): Gated Graph Neural Network (GGNN) is a type of recurrent neural network designed for graph data. It uses gated recurrent units (GRUs) to update node states in an iterative fashion and can be applied to graphs with different structures and sizes.

- Node2Vec (Grover & Leskovec, 2016): Node2Vec is another graph embedding algorithm that generates node embeddings using a biased random walk approach. It introduces two parameters that control the random walk process: the return parameter, which determines the likelihood of returning to the previous node, and the input parameter, which balances between breadth-first and depth-first exploration.

- Graph Convolutional Networks (GCN) (Kipf & Welling, 2016): GCN is a popular deep learning-based model for graph data that uses convolutional neural networks to learn node embeddings. GCN operates on a fixed graph structure and uses the same weights across all nodes to enable efficient computation.

- GAT (Veličković et al., 2017): Graph Attention Networks (GAT) is a deep learning-based model for graph data that uses attention mechanisms to learn node embeddings. GAT computes attention coefficients based on the features of nodes and their neighbors and aggregates information from neighboring nodes using a weighted sum.

- GraphSAGE (Hamilton et al., 2017): GraphSAGE is a variant of the GCN model that generalizes well to graphs with different structures and sizes. It learns node embeddings by aggregating information from the local neighborhood of each node using a trainable function. Unlike GCN, GraphSAGE places more emphasis on the individual nodes themselves, enabling it to better capture the diversity and complexity of real-world graph data.

- TAGCN (Du et al., 2017): Topology Adaptive Graph Convolutional Networks (TAGCN) is a graph embedding model that extends the GCN model by incorporating adaptive weights into the convolutional operation. These adaptive weights are learned from the graph topology and enable TAGCN to adjust its parameters based on the degree of each node, allowing it to better capture the local structure of the graph. This makes TAGCN more effective at modeling complex networks with diverse degrees and enhances its performance on tasks such as node classification, link prediction, and clustering.

- APPNP (Gasteiger et al., 2018): Approximate Personalized Propagation of Neural Predictions (APPNP) is a semi-supervised learning algorithm for graph data that improves upon the propagation scheme of GCN by incorporating ideas from personalized PageRank. APPNP uses a modified version of PageRank to propagate node representations across the graph and make predictions, enabling it to capture local and global information in the network and achieve improved performance on downstream tasks.

- Cluster-GCN (Chiang et al., 2019): Cluster-GCN is a scalable variant of the GCN model that partitions large graphs into clusters and performs GCN operations on each cluster separately. It allows for efficient parallelization and can handle much larger graphs than traditional GCN models.

- DAGNN (Liu et al., 2020): DAGNN is a deep learning model that operates on directed acyclic graphs (DAGs), outperforming conventional GNNs by leveraging the partial order as strong inductive bias besides other suitable architectural features.

- GATv2 (Brody et al., 2021): GATv2 is an improved version of the GAT model that addresses the static attention problem of the standard GAT layer by allowing every node to attend to any other node, unlike in the standard GAT where the ranking of attended nodes is unconditioned on the query node.

## I.3 HYPERPARAMETER SETTINGS

In this section, we present a list of the hyperparameters and final hyperparameters used in each baseline model for Ethereum link prediction. To ensure consistency across all models, we set the output node embedding size to 128 and the learning rate to 0.001.

**Random-walk based model settings.** We refer to the original literature of DeepWalk (Perozzi et al., 2014) and Node2Vec (Grover & Leskovec, 2016) to guide the hyperparameters tuning. We list the hyperparameters we tried in Table VIII.

Table VIII: Random-walk based models' hyperparameters for Ethereum link prediction.

| Model | Walk Length | Num of Walks | Window Size | $p$ | $q$ |
|---|---|---|---|---|---|
| DeepWalk | [20, 40, 50, 60, 70, 80] | [10, 20, 30, 40] | [3, 5, 10] | N.A. | N.A. |
| Node2Vec | [20, 40, 50, 60, 70, 80] | [10, 20, 30, 40] | [3, 5, 10] | [0.25, 0.5, 1] | [0.5, 1, 2] |

Through experimentation, we have found that a walk length of 20 and a window size of 5, with 40 walks per node, produce optimal results for DeepWalk. For our implementation of Node2Vec, we set hyperparameters $p = 0.5$ and $q = 2$, while keeping all other hyperparameters the same as for DeepWalk. These hyperparameter settings guide the random walk to explore the graph more efficiently, resulting in superior performance.

**Baseline GNN model settings.** Following the suggestions from the recent literature, we employ a three-layer architecture for all graph neural network (GNN) models, which allows us to capture sufficient information from the graph while preventing overfitting. The optimization of the models' hyperparameters is accomplished through a comprehensive grid-search strategy, drawing guidance from the respective original papers and code repositories of GGNN (Li et al., 2015), GCN (Kipf & Welling, 2016), GAT (Veličković et al., 2017), GraphSage (Hamilton et al., 2017), TAGCN (Du et al., 2017), APPNP (Gasteiger et al., 2018), Cluster-GCN (Chiang et al., 2019), DAGNN (Liu et al., 2020), and GATv2 (Brody et al., 2021). We list all parameters we implemented on each GNN model in Table IX.

Table IX: GNN models' hyperparameters for Ethereum link prediction.

| Model | Hidden Dimensions | Dropout Rate | Num of Heads | Negative Slope | $k$ |
|---|---|---|---|---|---|
| GGNN | [16, 32, 64, 128] | [0.1, 0.2, 0.5] | N.A. | N.A. | [2, 3, 5] |
| GCN | [16, 32, 64, 128] | [0.1, 0.2, 0.5] | N.A. | N.A. | N.A. |
| GAT | [16, 32, 64, 128] | [0.1, 0.2, 0.5] | [3, 4, 5] | N.A. | N.A. |
| GraphSage | [16, 32, 64, 128] | [0.1, 0.2, 0.5] | N.A. | N.A. | N.A. |
| TAGCN | [16, 32, 64, 128] | [0.1, 0.2, 0.5] | N.A. | N.A. | N.A. |
| APPNP | [16, 32, 64, 128] | [0.1, 0.2, 0.5] | N.A. | N.A. | [2, 3, 5] |
| Cluster-GCN | [16, 32, 64, 128] | [0.1, 0.2, 0.5] | N.A. | N.A. | N.A. |
| DAGNN | [16, 32, 64, 128] | [0.1, 0.2, 0.5] | N.A. | N.A. | [1, 2, 3] |
| GATv2 | [16, 32, 64, 128] | [0.1, 0.2, 0.5] | [3, 4, 5] | [0.01, 0.2, 0.5] | N.A. |

After conducting experiments, we have determined that all GNN models perform optimally with a hidden dimension 128 and a dropout rate 0.1. Another model DAGNN, while sharing the same parameters as GCN, adopts $k = 1$, where $k$ represents the number of hops visible by each node during training. For GAT and GATv2 Models, we set the number of heads to 3. The GATv2 model further includes a negative slope parameter set at 0.2, which is often used in deep learning models to address the "dying ReLU" problem by allowing for non-zero gradients even when the input is negative (Lu et al., 2019).

For the GGNN model, we set $k = 5$, which represents the number of steps executed during each iteration of the algorithm. These steps are used to update the hidden state of each node in the graph based on its previous state and the information from neighboring nodes.

On the other hand, for the APPNP model, a propagation step count (also $k$) of 3 is used. This parameter is used to dynamically control the amount of information that propagates across the graph during each iteration of the algorithm when computing personalized page rank. A larger $k$ will result in more information propagating across the graph and potentially larger improvements in performance, but also increased computational complexity.

## I.4 ETHEREUM LINK PREDICTION FOR NON-MATCHING ETHEREUM ADDRESSES

Ethereum link prediction can help monitor transaction trends of Ethereum. However, previous works only research on Ethereum dataset (Lin et al., 2021) (Lin et al., 2022). Therefore, we aim to compare

the performance between scenarios: one with the addition of X features and one without, especially for those Ethereum addresses without X matching. We employ five baseline models: GCN (Kipf & Welling, 2016), GAT (Veličković et al., 2017), APPNP (Gasteiger et al., 2018), TAGCN (Du et al., 2017), Cluster-GCN (Chiang et al., 2019).

**Implementation details.** Firstly, We divide all Ethereum graph edges into training, validation, and test sets, following a 70/10/20 ratio based on edge timestamps. We then remove the validation and test edges and choose the largest connected subgraph from the remaining edges to constitute our training graph. This process ensures that we only use old links to predict new ones. Next, we random choose the same number of edges as in Section 4.2 as positive training edges. And we make sure these edges do not contain Ethereum addresses which have matching X accounts. For balance, we generate an equal number of negative training edges by selecting a non-existent link from each Ethereum address within the positive training edges. Again, we make sure these negative training edges do not contain Ethereum addresses which have matching X accounts.

For the validation and test sets, we choose positive edges from the previously partitioned validation and test sets and ensure that both nodes in a validation or test edge are also present in the training graph. Also, positive validation and test edges do not contain any Ethereum addresses with matching X accounts.

For every positive edge within the validation and test sets, we randomly choose a non-existent link stemming from the same Ethereum address. Again, we make sure these negative validation and test edges do not contain Ethereum addresses which have matching X accounts. With this framework, we execute two experiments: one that incorporates X features and one that does not. Our goal is to compare the results of these two scenarios to determine the influence of X data on the analysis of Ethereum activities.

**Evaluation metrics.** We use the following four widely used metrics (Chen et al., 2020), to evaluate the performance of experiments in a comprehensive way: **(1) AUC-ROC**, **(2) Precision**, **(3) Recall**, and **(4) F1-score.** Note that, for all the experiments we repeat 5 times and report the average performance with standard deviation.

Table X: Results for Ethereum link prediction, with ↑, -, and ↓ indicating performance increase, maintaining and decrease, respectively.

| Method | Without X Features | | | | With X Features | | | |
|---|---|---|---|---|---|---|---|---|
| | AUC-ROC | Precision | Recall | F1 | AUC-ROC | Precision | Recall | F1 |
| GCN | 0.84±0.01 | 0.76±0.01 | 0.75±0.01 | 0.75±0.00 | 0.84±0.00 (↑) | 0.76±0.00 (↑) | 0.75±0.02 (↓) | 0.76±0.01 (↑) |
| GAT | 0.78±0.02 | 0.72±0.01 | 0.68±0.03 | 0.70±0.02 | 0.85±0.01 (↑) | 0.76±0.00 (↑) | 0.79±0.02 (↑) | 0.77±0.01 (↑) |
| TAGCN | 0.84±0.00 | 0.81±0.00 | 0.71±0.00 | 0.76±0.00 | 0.85±0.00 (↑) | 0.80±0.01 (↓) | 0.74±0.01 (↑) | 0.77±0.00 (↑) |
| Cluster-GCN | 0.88±0.00 | 0.81±0.00 | 0.78±0.01 | 0.80±0.01 | 0.90±0.00 (↑) | 0.82±0.00 (↑) | 0.82±0.01 (↑) | 0.82±0.00 (↑) |
| APPNP | 0.84±0.01 | 0.80±0.00 | 0.70±0.01 | 0.74±0.00 | 0.87±0.00 (↑) | 0.79±0.00 (↓) | 0.77±0.00 (↑) | 0.78±0.00 (↑) |

**Discussion.** Results shown in Table X demonstrate the superiority of GNN that leverage both graph structures and X features compared to those that rely solely on graph structures. Notably, incorporating features from matched X accounts significantly improves the performance of non-matching Ethereum addresses' link prediction across all models. For instance, GAT exhibits a substantial 7% increase in AUC-ROC when integrating X features, while APPNP shows an 7% surge in recall. This enhancement is attributable to the unique insights obtained from X data, which reveal behaviors and characteristics of Ethereum addresses not discernible from Ethereum transaction data alone. Additionally, incorporating X data enables an effective utilization of the message passing mechanism in GNNs in the dataset. These findings underscore the potential of our dataset, which benefits analysis of not only Ethereum addresses with matching X accounts but also those Ethereum addresses lacking external information.

## J   DETECTION OF WASH-TRADING ADDRESSES

### J.1   FEATURE ENGINEERING

In our analysis for wash-trading address detection, we incorporate a set of eight structural features identified in our statistical analysis section. These structural features act as our primary input for Ethereum addresses.

For DeepWalk and Node2Vec, we explore the effectiveness of features generated through random walks alone as well as a combination of random-walk features and semantic features derived from associated X accounts. This approach enables us to evaluate the impact of incorporating off-chain information on the models' performance.

For other Graph Neural Network (GNN) models, we similarly compare the performance when using only the structural features against a combined set of structural and semantic features, with the semantic features being extracted from the respective X accounts. The aim here is to ascertain the extent to which the inclusion of off-chain social information can improve the accuracy of wash-trading address detection.

The node embeddings, which serve as a condensed representation of the nodes, are derived from the trained models. It's worth noting that, for each model and experimental setting, we report the average results over five experiments, along with the associated standard deviations.

### J.2   MODEL DESCRIPTIONS

In this section, we introduce the models that we utilized as baselines for detecting wash-trading Ethereum addresses. All of the models were already introduced in Section I.2.

### J.3   HYPERPARAMETER SETTINGS

In this section, we present a list of the hyperparameters and final hyperparameters used in each baseline model for wash-trading addresses detection. To ensure consistency across all models, we set the dimension of the output node embeddings to 2 as there are two classes of addresses: wash-trading address and normal address. And we set the learning rate to 0.001.

**Random-walk based models settings.** We refer to the original literature of DeepWalk (Perozzi et al., 2014) and Node2Vec (Grover & Leskovec, 2016) to guide the hyperparameters tuning. We list the hyperparameters we tried in Table XI.

Table XI: Random-walk based models' hyperparameters for wash-trading addresses detection.

| Model | Walk Length | Num of Walks | Window Size | $p$ | $q$ |
|---|---|---|---|---|---|
| DeepWalk | [20, 40, 80] | [10, 20, 30, 40, 50] | [3, 5, 10] | N.A. | N.A. |
| Node2Vec | [20, 40, 80] | [10, 20, 30, 40, 50] | [3, 5, 10] | [0.25, 0.5, 1] | [0.5, 1, 2] |

Through experimentation, we have determined that a walk length of 20 and a window size of 5, with 40 walks per node, produce optimal results for DeepWalk. For our implementation of Node2Vec, we set hyperparameter $p = 0.5$ and $q = 2$, while keeping all other hyperparameters the same as those used in DeepWalk.

**Baseline GNN model settings.** Following the suggestions from the recent literature, we employ a three-layer architecture for all graph neural network (GNN) models, which allows us to capture sufficient information from the graph while preventing overfitting. We employ a grid-search strategy, informed by the recommendations and code repositories of several baseline models, including GGNN (Li et al., 2015), GCN (Kipf & Welling, 2016), GAT (Veličković et al., 2017), GraphSage (Hamilton et al., 2017), APPNP (Gasteiger et al., 2018), and GATv2 (Brody et al., 2021). We list the hyperparameters in the Table XII.

After experimenting, we have decided that all GNN models share a hidden dimension 128, and a dropout rate 0.1. For GAT and GATv2 Models, we set the number of heads to 3. The GATv2 model further includes a negative slope parameter set at 0.2, which is often used in deep learning models

Table XII: GNN models' hyperparameters for Ethereum link prediction.

| Model | Hidden Dimensions | Dropout Rate | Num of Heads | Negative Slope | $k$ |
|---|---|---|---|---|---|
| GGNN | [16, 32, 64, 128] | [0.1, 0.2, 0.5] | N.A. | N.A. | [2, 3, 5] |
| GCN | [16, 32, 64, 128] | [0.1, 0.2, 0.5] | N.A. | N.A. | N.A. |
| GAT | [16, 32, 64, 128] | [0.1, 0.2, 0.5] | [3, 4, 5] | N.A. | N.A. |
| GraphSage | [16, 32, 64, 128] | [0.1, 0.2, 0.5] | N.A. | N.A. | N.A. |
| TAGCN | [16, 32, 64, 128] | [0.1, 0.2, 0.5] | N.A. | N.A. | N.A. |
| APPNP | [16, 32, 64, 128] | [0.1, 0.2, 0.5] | N.A. | N.A. | [2, 3, 5] |
| Cluster-GCN | [16, 32, 64, 128] | [0.1, 0.2, 0.5] | N.A. | N.A. | N.A. |
| DAGNN | [16, 32, 64, 128] | [0.1, 0.2, 0.5] | N.A. | N.A. | [1, 2, 3] |
| GATv2 | [16, 32, 64, 128] | [0.1, 0.2, 0.5] | [3, 4, 5] | [0.01, 0.2, 0.5] | N.A. |

to address the "dying ReLU" problem by allowing for non-zero gradients even when the input is negative (Lu et al., 2019).

For the GGNN model, we set $k = 5$, which represents the number of steps executed during each iteration of the algorithm. These steps are used to update the hidden state of each node in the graph based on its previous state and the information from neighboring nodes.

For GAT and GATv2 Models, we set the number of heads to 3. The GATv2 model further includes a negative slope parameter set at 0.2.

On the other hand, for the APPNP model, a propagation step count (also $k$) of 3 is used. This parameter is used to dynamically control the amount of information that propagates across the graph during each iteration of the algorithm when computing personalized page rank. A larger $k$ will result in more information propagating across the graph and potentially larger improvements in performance, but also increased computational complexity.

Overall, the tuning of these and additional parameters is crucial for achieving robustness and accuracy in the detection of wash-trading Ethereum addresses.

# K   MATCHING LINK PREDICTION

## K.1   FEATURE ENGINEERING

Feature engineering plays a crucial role in our analysis, allowing us to distill complex information into a tractable format suitable for our machine learning models. For Ethereum addresses, we extract a set of structural features. These structural features encapsulate transaction characteristics of Ethereum addresses, capturing essential aspects of their on-chain activities.

When it comes to X accounts matched with Ethereum addresses, we supplement the same structural features as Ethereum addresses with additional semantic features, extracted from the X account data. The semantic features add a rich layer of off-chain information, capturing nuanced aspects of users' online social behavior.

For X accounts that do not have a matched Ethereum address, we preserve the dimensionality of the feature space by appending an eight-dimensional zero-vector to the existing structural features. This maintains consistency in the input feature size across all nodes, thus making it suitable for the training of GNN models. As a result, all nodes in this training share an input vector of 16 dimensions.

After training, we extract node embeddings from our models. Each embedding is a compact representation of a node, encapsulating the node's position and connectivity within the network. For an edge linking two nodes, we generate the edge embedding by concatenating the respective two node embeddings. This resultant edge embedding carries combined information about both linked nodes, providing a comprehensive feature set for edge-level analysis.

### K.2 Model Descriptions

In this section, we introduce the models that we utilized as baselines for prediction matching links between Ethereum addresses and X accounts. As some of the models were already introduced in Section I.2, we will only introduce the new models used in this experiment.

- HIN2Vec (Fu et al., 2017): HIN2Vec is a graph embedding model designed specifically for heterogeneous information networks (HINs). It uses a combination of meta-path-based random walks and skip-gram models to learn distributed representations of nodes in a HIN.
- R-GCN (Schlichtkrull et al., 2018): Relational Graph Convolutional Networks (R-GCN) is a variant of the GCN model that can operate on multi-relational graphs with varying edge types. It uses different weight matrices for each relation in the graph to enable edge-type-specific feature transformation and learning.

### K.3 Hyperparameter Settings

For consistency, we maintain the same hyperparameter setting for this matching link prediction as used in our Ethereum link prediction model for all the models presented in this study. In particular, all models share a common output node embedding size of 128, which determines the size of the learned representation for each node in the graph. Additionally, a learning rate of 0.001 is used across all models to control the rate of learning during training. These hyperparameters were chosen based on experimentation and validation performance on the Ethereum dataset, and they are kept consistent throughout our analysis for fair comparison of the different models.

For the random-walk based graph embedding models DeepWalk (Perozzi et al., 2014), Node2Vec (Grover & Leskovec, 2016), and HIN2Vec (Fu et al., 2017), we set a walk length of 20 and a window size of 5, denoting the length of the random walk sequences and the window of context for prediction, respectively. Additionally, we set 40 walks per node to ensure adequate sampling of the graph. Specifically for Node2Vec, we set $p = 0.5$ and $q = 2$, respectively, controlling the random walk's likelihood to revisit a node and explore undiscovered parts of the graph.

For our GNN-based models, we configure them with three layers, a hidden layer dimension of 128, and a dropout rate of 0.1. R-GCN, while sharing these parameters, distinctly incorporates three types of relational edges to capture the heterogeneity of our constructed graph.

The DAGNN model, keeping the GCN parameters consistent, specifically uses a hyperparameter $k = 1$. This denotes the number of hops each node perceives during training. With $k = 1$, the model takes into account only the immediate neighbors of each node for updating its representation.

For the GGNN model, we adopt a setting of $k = 5$, representing the steps taken during each iteration. These steps facilitate the update of each node's hidden state based on prior states and input from adjacent nodes.

For both the GAT and GATv2 models, we determine the number of heads to be 3. Specifically, the GATv2 model incorporates an additional negative slope parameter, set at 0.2.

Conversely, for the APPNP model, we use a propagation step count (denoted as $k$) of 3. This parameter dynamically dictates the volume of information flowing across the graph during each iteration when computing the personalized page rank. A higher $k$ can potentially enhance performance due to increased information flow but comes at the expense of heightened computational demand.

## L URL to Dataset

We deploy our dataset on a GitHub repository: `https://github.com/Persdre/EX-Graph`.

## M Dataset Documentation

The EX-Graph dataset is a collection of on-chain Ethereum transaction records and off-chain X following-follower data, arranged as graphs in DGL (Deep Graph Library) format. The dataset

is suitable for various research tasks, including Ethereum link prediction, wash-trading addresses detection, and matching link prediction.

## M.1 INTENDED USE

This dataset is designed for researchers and developers who are interested in exploring the behavior of Ethereum transactions from both on-chain and off-chain domains, as well as the relationships between Ethereum addresses and X users. The dataset can facilitate research studies on various topics, including fraud detection, transaction analysis, and social network analysis.

## M.2 ACCESSING THE DATA

Due to its large size, the EX-Graph dataset is hosted on Google Drive for easy access and download. The dataset comprises these main parts:

- **X Graph**: Represents X accounts and their relationships.
- **Ethereum Graph**: Describes Ethereum addresses and their transaction history.
- **Ethereum Graph with X features**: A DGL graph that integrates Ethereum addresses, their transactions, and features from both Ethereum and matched X accounts. Used for Ethereum link prediction tasks.
- **Wash-trading Addresses Detection Training Graph**: The training graph used for detecting wash-trading addresses.
- **Wash-trading Addresses Detection Validation Graph**: The validation graph for the same task.
- **Wash-trading Addresses Detection Test Graph**: The test graph for the task.
- **Matching Link Prediction Graph**: A DGL graph utilized for the matching link prediction task.

To access the data, please follow the links provided in the GitHub README file and download the relevant files from Google Drive.

## M.3 CODE FOLDERS

The repository contains three code folders that correspond to the different use cases of the EX-Graph dataset:

- Ethereum link prediction: This folder contains the source code for using the dataset for Ethereum link prediction.
- Wash trading address detection: This folder contains the source code for using the dataset for detecting wash-trading addresses.
- Matching link prediction: This folder contains the source code for using the dataset for matching link prediction.

## M.4 DATA FILES

In addition to the code folders, the repository includes two data files:

- X handles (converted to numerical id for anonymization) with their matching Ethereum addresses from OpenSea: This file contains the X handles of various users and their corresponding Ethereum addresses.
- Wash-trading transaction records collected from Dune: This file contains records of wash-trading transactions collected from Dune [13].

---

[13]https://dune.com/queries/364051

## M.5 LICENSE

The EX-Graph dataset is released under the Creative Commons Attribution-NonCommercial-ShareAlike (CC BY-NC-SA) license. This means that anyone can use, distribute, and modify the data for non-commercial purposes as long as they give proper attribution and share the derivative works under the same license terms.

## N AUTHOR STATEMENT

As authors of the EX-Graph dataset, we hereby declare that we assume full responsibility for any liability or infringement of third-party rights that may come up from the use of our data. We confirm that we have obtained all necessary permissions and/or licenses needed to share this data with others for their own use. In doing so, we agree to indemnify and hold harmless any person or entity that may suffer damages resulting from our actions.

Furthermore, we confirm that our EX-Graph dataset is released under the Creative Commons Attribution-NonCommercial-ShareAlike (CC BY-NC-SA) license. This license allows anyone to use, distribute, and modify our data for non-commercial purposes as long as they give proper attribution and share the derivative works under the same license terms. We believe that this licensing model aligns with our goal of promoting open access to high-quality data while respecting the intellectual property rights of all parties involved.

## O HOSTING PLAN

After careful consideration, we have chosen to host our code and data on GitHub. In addition, we deploy a website as a leaderboard for researchers to submit their experiment results on the EX-Graph we propose. Our decision is based on various factors, including the platform's ease of use, cost-effectiveness, and scalability. We understand that accessibility is key when it comes to data management, which is why we will ensure that our data is easily accessible through a curated interface. We also recognize the importance of maintaining the platform's stability and functionality, and as such, we will provide the necessary maintenance to ensure that it remains up-to-date, bug-free, and running smoothly.

At the heart of our project is the belief in open access to data, and we are committed to making our data available to those who need it. As part of this commitment, we will be updating our GitHub repository regularly, so that users can rely on timely access to the most current information. We hope that by using GitHub as our hosting platform, we can provide a user-friendly and reliable solution for sharing our data with others.

