# ETGraph: A Pioneering Dataset Bridging Ethereum and Twitter

## Abstract

While numerous public blockchain datasets are available, their utility is constrained by a singular focus on blockchain data. This constraint limits the incorporation of relevant social network data into blockchain analysis, thereby diminishing the breadth and depth of insight that can be derived. To address the above limitation, we introduce ETGraph, a novel dataset that authentically links Ethereum and Twitter, marking the first and largest dataset of its kind. ETGraph combines Ethereum transaction records (2 million nodes and 30 million edges) and Twitter following data (1 million nodes and 3 million edges), bonding 30,667 Ethereum addresses with verified Twitter accounts sourced from OpenSea. Detailed statistical analysis on ETGraph highlights the structural differences between Twitter-matched and non-Twitter-matched Ethereum addresses. Extensive experiments, including Ethereum link prediction, wash-trading Ethereum addresses detection, and Twitter-Ethereum matching link prediction, emphasize the significant role of Twitter data in enhancing Ethereum analysis. ETGraph is available at `https://etgraph.deno.dev/`.

## 1 Introduction

Blockchain technology is known for its secure, unchangeable, and anonymous records (Gao et al., 2018). Among various blockchains, Ethereum is one of the largest and has shown substantial growth. Its cryptocurrency ETH market value hit 19 billion dollars as of September 23, 2023 [1]. Another example is Non-Fungible Tokens (NFTs), deployed on Ethereum, attracting numerous celebrities to the blockchain (Nadini et al., 2021). However, the anonymity inherent to blockchain has drawn various financial crimes such as phishing scams and wash trading, particularly within Ethereum, making research on Ethereum necessary (Dyson et al., 2019; Li et al., 2020). Several studies released public Ethereum datasets to facilitate research in this area (Lin et al., 2022; Shamsi et al., 2022). In analyzing activities within Ethereum, researchers are increasingly employing graph analysis, specifically Graph Neural Network (GNN) methods (Liu et al., 2022; Wang et al., 2022).

However, prior studies have failed to address a notable issue: anonymous Ethereum addresses are hash strings which do not contain additional features. This absence of features limits the efficacy of GNN models. Two methods exist to get node features: manually designed features and network representation learning features (Li et al., 2022). However, both types are derived from the Ethereum graph's structure. Recent research indicated that the interaction between features and structures substantially contributes to the degradation of GNN performance (Yang et al., 2022). We note that some Twitter users utilize the Ethereum Name Service (ENS) for their usernames, associating a human-readable string with a specific Ethereum address. This connection indicates that the Ethereum address and the Twitter account belong to the same entity. One existing work (Béres et al., 2021) enhanced Ethereum address features by linking 890 Twitter users to addresses using this ENS format.

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

 Twitter accounts. Furthermore, we obtain embeddings for all Twitter accounts using the DeepWalk algorithm in the Twitter graph (Yuan et al., 2020). Drawing from the methodology articulated in (Chen et al., 2020), we select a set of eight structural features that holistically represent nodes within the Ethereum transaction graph inspired by these insights. These features are degree, in-degree, out-degree, neighbors, in-neighbors, out-neighbors, the maximum number of transactions with neighbors, and average neighbor degree. An in-depth description of these features are in Appendix C.

We leverage BERT (Devlin et al., 2018) for semantic feature extraction, converting the text profile $s_n$ of each Twitter account $n$'s text profile $s_n$ into a 768-dimensional embedding $h_n$ :

$$h_n = \text{BERT}(s_n), h_n \in \mathbb{R}^{768}, \tag{1}$$

The reason behind choosing BERT is in our Appendix C.

Then, we enhance this semantic representation by appending the logarithm of the number of followers and followings, resulting in a 770-dimensional embedding $h'_n \in \mathbb{R}^{770}$ :

$$h'_n = h_n \oplus \left(\log(\# \text{ followers} + 1), \log(\# \text{ followings} + 1)\right), h'_n \in \mathbb{R}^{770}, \tag{2}$$

This extended feature set is then processed. Each node in the Ethereum transaction and Twitter following graph is equipped with an 8-dimensional structural feature set. Furthermore, every matched Twitter account now possesses a 770-dimensional semantic feature set. To maintain uniformity across diverse feature sets and ensure a balanced model influence, we implement Principal Component Analysis (PCA). This adjustment effectively reduces the dimensionality of the Twitter semantic features to 8, denoted as $h''_n \in \mathbb{R}^8$ :

$$h''_n = \text{PCA}(h'_n), h''_n \in \mathbb{R}^8. \tag{3}$$

## 4 STATISTICS AND EXPERIMENTS

In this section, we first do an in-depth statistical analysis of our carefully curated ETGraph dataset in Section 4.1. The focus of this analysis is the structural features mentioned in Section 3.3 of Ethereum addresses that are matched with Twitter accounts versus those that are not, as well as differentiating wash-trading Ethereum addresses from non-wash-trading ones (with the latter being treated as normal Ethereum addresses). Then we conduct comprehensive experiments, aiming to address the following three research questions:

*Q1:* **To what extent does Twitter auxiliary information improve the performance of link prediction in Ethereum transactions?**

*Q2:* **How significantly does Twitter auxiliary information enhance the detection of wash-trading addresses in Ethereum transactions?**

*Q3:* **Is it feasible to predict additional matching links based on known matching links?**

### 4.1 ETGRAPH STATISTICAL ANALYSIS

#### 4.1.1 OVERALL ANALYSIS

Table 2: Summary of ETGraph. # Wash-trading Addresses listed in the Twitter row means how many wash-trading addresses match to twitter accounts.

| Graph | # Nodes | # Edges | Avg. Degree | # Matched Accounts | # Wash-trading Addresses |
|---|---|---|---|---|---|
| Ethereum | 2,610,465 | 29,585,858 | 11.33 | 30,667 | 1,445 |
| Twitter | 1,103,509 | 3,768,281 | 3.42 | 30,667 | 3 |

ETGraph combines the Ethereum and Twitter graphs, as shown in Table 2. Ethereum has a higher average degree compared with Twitter graph. However, the quantity of Ethereum addresses that match with a Twitter account is notably small. The limited wash-trading addresses in the dataset highlight data imbalance, which is often seen in financial data. Only three wash-trading addresses match with Twitter, reflecting the hidden nature of wash-trading activities. Overall, ETGraph gives a glimpse into Ethereum transactions and wash-trading behaviors, highlighting the difficulty in detecting wash-trading addresses in Ethereum.

### 4.1.2 ETHEREUM ADDRESSES BREAKDOWN

In terms of the eight structural features as mentioned in Section 3.3, in Table 3 we statistically compare the 30,667 Twitter-matched Ethereum addresses with the rest of the Ethereum addresses. Similarly, for Ethereum graph, we further compare the 1,445 wash-trading Ethereum addresses with the other normal addresses, as shown in Table 4.

Table 3: Comparison between Twitter-matched and non-Twitter-matched Addresses.

| Features | Twitter-matched Addresses | | | Non-Twitter-matched Addresses | | |
|---|---|---|---|---|---|---|
| | Average | Median | Std | Average | Median | Std |
| Degree | 28.26 | 2.00 | 492.39 | 12.98 | 2.00 | 80.75 |
| In-degree | 15.24 | 1.00 | 309.76 | 6.48 | 1.00 | 38.55 |
| Out-degree | 13.03 | 1.00 | 185.41 | 6.50 | 1.00 | 51.65 |
| Neighbors | 26.58 | 2.00 | 402.22 | 12.64 | 2.00 | 73.85 |
| In-neighbors | 15.24 | 1.00 | 309.76 | 6.48 | 1.00 | 38.55 |
| Out-neighbors | 13.03 | 1.00 | 185.41 | 6.50 | 1.00 | 51.65 |
| Max. txns with neighbors | 6.78 | 3.00 | 27.94 | 2.89 | 1.00 | 30.42 |
| Average neighbor degree | 33076.00 | 31105.75 | 27921.99 | 1442.87 | 149.00 | 7170.93 |

Table 4: Comparison between wash-trading and normal addresses on Ethereum graph.

| Features | Wash-Trading Addresses | | | Normal Addresses | | |
|---|---|---|---|---|---|---|
| | Average | Median | Std | Average | Median | Std |
| Degree | 137.61 | 15.00 | 377.70 | 13.02 | 2.00 | 92.74 |
| In-degree | 50.53 | 7.00 | 115.58 | 6.52 | 1.00 | 48.46 |
| Out-degree | 87.09 | 6.00 | 283.29 | 6.50 | 1.00 | 53.83 |
| Neighbors | 134.66 | 14.00 | 371.35 | 12.67 | 2.00 | 82.41 |
| In-neighbors | 87.09 | 6.00 | 283.29 | 6.50 | 1.00 | 53.83 |
| Out-neighbors | 50.53 | 7.00 | 115.58 | 6.52 | 1.00 | 48.46 |
| Max. txns with neighbors | 18.38 | 8.00 | 55.60 | 2.91 | 1.00 | 30.37 |
| Average neighbor degree | 423.95 | 247.84 | 986.01 | 1740.28 | 151.71 | 8220.95 |

From Tables 3 and 4, it is clear that Twitter-matched and wash-trading addresses show higher activity levels across the first seven features compared to others. The increased in-degree, out-degree, and maximum transactions with neighbors suggest Twitter-matched addresses are part of an active sub-network within Ethereum. Wash-trading addresses' higher degree, and number of neighbors underscore their active transaction behavior. These addresses exhibit higher median in-degree, out-degree values, and a larger average degree, indicating significant transaction volumes.

The significant average neighbor degree for Twitter-matched addresses primarily reflects frequent transactions with high-degree nodes, such as smart contracts in activities like NFT minting (Wang et al., 2021). Contrastingly, wash-trading addresses exhibit a lower average neighbor degree, likely due to their circular transactions among a confined group of addresses (Victor & Weintraud, 2021) which limits interaction with the wider Ethereum network's high-degree nodes. We provide a more detailed analysis in Appendix H.

In conclusion, we highlight distinct disparities in transaction activities and network centrality between Twitter-matched and non-Twitter-matched Ethereum addresses, as well as between wash-trading and normal addresses. In the following experiments, we explore the correlation between these structural features and Ethereum transaction link prediction, with special focus on the classification of Ethereum addresses, especially wash-trading addresses.

### 4.2 ETHEREUM LINK PREDICTION (*Q1*)

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

**Discussion.** Results shown in Table 5 demonstrate the superiority of deep learning models that leverage both graph structures and Twitter features compared to those that rely solely on graph structures. Notably, incorporating features from matched Twitter accounts significantly improves the performance of all models, as evidenced by enhanced metrics. For instance, TAGCN exhibits a substantial 7% increase in AUC-ROC when integrating Twitter features, while DAGNN shows an 8% increase in AUC-ROC and a remarkable 10% surge in recall. This improvement is attributed to the fact that the Ethereum transaction graph primarily captures structural relationships among Ethereum addresses, lacking extensive descriptive or attribute data about individual nodes. By incorporating Twitter

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

## 5 CONCLUSION

We present ETGraph, a pioneering and extensive dataset that bridges Ethereum and Twitter. Experiments on ETGraph highlights integrating additional Twitter data significantly boosts the effectiveness of various tasks on Ethereum. Our research augments the understanding of the interplay between on-chain and off-chain worlds and paves the way for novel methods in probing on-chain activities using off-chain supplementary data. However, we recognize that there are still areas to work on, such as collecting more matching links between on-chain and off-chain graphs, and wash-trading Ethereum addresses to counter the data imbalance. Our next steps involve incorporating these enhancements and constantly updating our graph with more on-chain and off-chain nodes and edges. Additionally, inspired by this research, we aspire to enhance graph learning by disseminating diverse field data, particularly for real-world feature-sparse datasets.