# OpenReview forum: "EX-Graph: A Pioneering Dataset Bridging Ethereum and X"
_ICLR.cc/2024/Conference — ICLR 2024 poster_

### Official Review · Reviewer_hHyG · 2023-10-24

**Soundness:** 3 good
**Presentation:** 3 good
**Contribution:** 3 good
**Rating:** 8
**Confidence:** 3

**Summary:**

The paper contributes a dataset, termed ETGraph, that links Ethereum addresses with Twitter accounts. The dataset is structured in the form of a heterogeneous graph and inlcudes links between pairs of Ethereum addresses that transact with each other, links between Twitter accounts that follow/are followed by each other and links between Ethereum addresses and Twitter accounts that have been matched to be owned by the same (typically, physical) entity/person. This information has been scraped from opensea and the public Twitter API. The dataset also includes information about (a relatively small number of) money-laundering addresses.

In addition, the paper also evaluates the dataset over a series of benchmark Graph Neural Network (GNN) models. Specifically, it compares whether the new dataset (with Twitter data) improves the ability of the models to predict Q1) links between Ethereum addresses, i.e., transactions, Q2) money laundering addresses, and Q3) matches between Ethereum addresses and Twitter accounts. With few exceptions, most evaluations suggest improved performance - at average levels around 6%-8% (maybe higher in some cases/lower in others) - with the new dataset in a couple of standard metrics including ROC, AUC, precision, recall and F1 scores.

**Strengths:**

- The dataset is novel and nicely connects Ethereum data to Twitter data. It involves a lot of hard work to put this dataset together and bring it to the heterogeneous graph structure. This is likely to have an impact (be useful) to researchers in the blockchain field.
- Regarding reproducibility, the paper is transparent and I could verify the links to the provided datasets and code.
- The evaluations indicate generally improved performance for a wide range of benchmark GNN models.

**Weaknesses:**

- The paper is not self-contained and a lot of things need to be assessed through the external links to the dataset or through the appendix. I am not very familiar with dataset papers, so maybe this is generally so.
- For this reason, I cannot comment on the ethical considerations about releasing such a dataset.
- Q2 in the evaluation seems to make no sense to me: since there are only 3 matched addresses in the dataset, why should we anticipate improved performance with this dataset? The results are, thus, questionable to me regarding Q2.
- The insight that Twitter-matched addresses are more active in Ethereum is rather expected since Twitter is known to be popular (as the authors also confirm) by main actors of the Ethereum ecosystem.
- On a subjective comment, I am not sure if ICLR is the right venue to maximize visibility for this type of datasets. However, Chartalist, a recent publication in NeurIPS about a related blockchain dataset seems to suggest the contrary. Also, related to my comment about ethical considerations above, the authors seem to have followed the related discussion on openreview about Charalist and have included statements in the appendix. However, I am not in a position to assess those, because I am not aware of the exact regulations regarding such datasets. For this reason, I raise a flag for ethics review below - but I would be happy to take it down if this is clarified.
- There are some technical things that I discuss below, but these seems to be more minor in nature. As an example, the set of features degree (3 features) and neighbor (3 features) seem to be highly linearly correlated (as also suggested by their statistics in Table 3). Keeping only of them, i.e., one from the "degree" family, and one from the "neighbor" family, would potentially improve the performance of the networks. I provide a couple of more minor comments below.

**Questions:**

- Can you please discuss the weaknesses above?
- Is overs-smoothing a relevant problem in the evaulations and if yes, have you taken steps to address it?

Minor:
- Figure 2: In the Feature Extraction column, we read in the first box "Degree"-> Structural Features. So, here, you mean that look at the degree of the node or that "Degree" is some kind of network like BERT and Deepwalk in the next two boxes of that column? I found this slightly confusing. Also, which structural features are extracted here since ETH addresses don't contain any further information.
- Below Figure 2: "The total collection accounts for 30,387 Twitter profiles". I was confused here: why is this lower than the initial number (since, according to the previous lines more data was added) and why are Followers and Friends useful here? Friends are not discussed elsewhere if I am not wrong.

Trivia:
- should it be Twitter or X :)?
- is it 19 billion or 190 billion as of September 23, the market value for Ethereum (page 1)?
- please define abbreviations the first time that they are used, e.g., AUC.

Note on my score: although I rated the paper as slightly above the acceptance threshold, I may decrease my score if my concerns are not addressed. I also hope that through the discussion, I will be able to increase my confidence.

**Post-rebuttal**: Based on the authors' response, I raised my score to 8. I cannot remove the ethics flag, but, I believe that the authors have addressed this concern. I will not increase my confidence, since datasets is not my exact area of expertise - although, I am fairly confident that this dataset will be useful to blockchain research.

**Details Of Ethics Concerns:**

The authors release a dataset in this paper containing matched Twitter accounts with Ethereum addresses. Whether this is allowed is outside my expertise and with this flag, I want to be sure that this is double-checked. Otherwise, the paper is written in a very transparent way and includes all details needed to verify its claims, including the dataset.

**Post rebuttal**: I appreciate the authors' response, and I think that it addresses the ethics concerns.

---

> ### Author Response · Authors · 2023-11-17
> **Response to Reviewer hHyG**
>
> Dear Reviewer hHyG,
>
> We are profoundly grateful for your comprehensive review and the acknowledgment of our work's contributions. Your appreciation of our dataset, ETGraph, which innovatively links Ethereum addresses to Twitter accounts, is highly valued. The effort to construct this novel and complex heterogeneous graph, as well as its anticipated impact on blockchain research, has been a significant part of our endeavor. We are particularly gratified that you recognized the transparency and reproducibility of our work, essential qualities we strive to maintain. Additionally, your observation of the improved performance across various GNN models using our dataset, as evidenced by enhanced metrics like ROC, AUC, precision, recall, and F1 scores, affirms the practical utility and effectiveness of our research. Your insightful feedback is not only a validation of our current efforts but also a strong encouragement for our future endeavors. Regarding the weaknesses you have pointed out, we will address each of them in detail below to provide further clarification and insight.
>
>
> 1. The paper is not self-contained and a lot of things need to be assessed through the external links to the dataset or through the appendix. I am not very familiar with dataset papers, so maybe this is generally so.
>
>    Thank you for your feedback regarding the self-contained nature of our paper. We understand your concern about the necessity to refer to external links and appendices for complete details. It is indeed common in dataset papers, especially those involving complex and extensive datasets like ours, to supplement the main text with additional resources. For example, [Huang et al., 2023] propose a Temporal Graph Benchmark for Machine Learning on Temporal Graphs, moving all experiment settings to the appendix and use an external website to contain many details. This approach helps maintain the clarity and conciseness of the paper while ensuring that detailed and technical aspects are readily accessible to those who seek more in-depth information.
>
>    Also, we acknowledge your concerns and have refined our paper by incorporating some key technical details, previously in the appendix into the main text. Notably, we have added an explanation for choosing BERT, now highlighted in blue in Section 3.3 of the revised paper.
>
>
> 2. For this reason, I cannot comment on the ethical considerations about releasing such a dataset.
>
>     We acknowledge and take seriously your concerns regarding the ethical implications associated with the release of our dataset. To address these concerns, we have detailed our ethical considerations and guidelines in Appendix D, focusing on the data collection procedures from OpenSea and Twitter. Throughout the development of our work, we have conscientiously adhered to Twitter's specific guidelines, notably those pertaining to off-Twitter matching and the redistribution of content for non-commercial research purposes. Recognizing the sensitive nature of personal data usage, we have implemented rigorous measures to ensure the utmost protection of user privacy. This includes the thorough anonymization of any data that could potentially be traced back to individual Twitter users, ensuring that our dataset is utilized exclusively for research objectives.
>
>     We acknowledge the importance of addressing potential misuse of our model, such as for intrusive surveillance. Our primary goal is to explore the synergy between on-chain and off-chain data for enhancing network analytics, strictly within an academic context. Our model is designed with the intention of analyzing Ethereum transaction patterns and protecting the interests of genuine users. We want to make it clear that our model does not support, nor is it designed for, any form of widespread surveillance.
>
>     Regarding the IRB review process, our study relies exclusively on publicly available and anonymized data, thus ensuring individual privacy and eliminating the risk of identifying specific entities. After consultation with the chair of our institution's IRB, we have been assured that our research complies with the necessary ethical guidelines and qualifies for exemption. This exemption is based on the criteria that the information we gathered cannot be used to ascertain the identities of subjects, either directly or indirectly (source: [IRB Northwestern University](https://irb.northwestern.edu/submitting-to-the-irb/types-of-reviews/exempt-review.html#:~:text=or%20reputation%3B%20OR-,The%20information%20obtained%20is%20recorded%20by%20the%20investigator%20in%20such,IRB%20conducts%20limited%20IRB%20review)).

---

> > ### Author Response · Authors · 2023-11-17
> >
> > 3. Q2 in the evaluation seems to make no sense to me: since there are only 3 matched addresses in the dataset, why should we anticipate improved performance with this dataset? The results are, thus, questionable to me regarding Q2.
> >
> >     Thank you for your observations regarding Q2 in our evaluation. We understand your concern about the seemingly limited number of matched addresses in the dataset and its impact on performance improvement. To clarify, our experiments extended beyond merely analyzing matched Ethereum addresses. In response to the reviewer AoG7's Point 3, we discovered that incorporating Twitter data enhances the link prediction for non-matching Ethereum addresses as well. This improvement is largely attributed to the message-passing mechanism inherent in Graph Neural Networks (GNN). Through this mechanism, semantic information from the few matched addresses can disseminate throughout the network, particularly influencing nodes directly connected to these matched addresses. Therefore, despite the small number of direct matches, the indirect influence of these connections via semantic information propagation would significantly contribute to the overall performance enhancement observed in our results.
> >
> >
> >
> > 4. The insight that Twitter-matched addresses are more active in Ethereum is rather expected since Twitter is known to be popular (as the authors also confirm) by main actors of the Ethereum ecosystem.
> >
> >     We appreciate your observation regarding the increased activity of Twitter-matched addresses in the Ethereum ecosystem. Indeed, while this may appear intuitive given Twitter's popularity among key Ethereum participants, we would like to highlight the absence of empirical studies that have previously explored and validated this correlation. Our research represents the first systematic effort to analyze and substantiate this hypothesis with concrete evidence. This undertaking not only confirms a widely held perception but also significantly enriches our understanding of the interplay between social media behavior and blockchain activity.
> >
> >     Furthermore, a principal contribution of our work is we propose a pioneering dataset that matches Twitter and Ethereum entities. This resource not only facilitates the verification of prevalent insights but also opens avenues for future explorations in the domain of blockchain and social media analytics.
> >
> >
> > 5. On a subjective comment, I am not sure if ICLR is the right venue to maximize visibility for this type of datasets. However, Chartalist, a recent publication in NeurIPS about a related blockchain dataset seems to suggest the contrary. Also, related to my comment about ethical considerations above, the authors seem to have followed the related discussion on openreview about Charalist and have included statements in the appendix. However, I am not in a position to assess those, because I am not aware of the exact regulations regarding such datasets. For this reason, I raise a flag for ethics review below - but I would be happy to take it down if this is clarified.
> >
> >     We thank you for your insights on both the suitability of ICLR as a venue for our dataset publication and your concerns regarding ethical considerations. While ICLR's expanding scope to include interdisciplinary research makes it a fitting platform for our blockchain dataset, as demonstrated by similar publications like Chartalist at NeurIPS, we appreciate your thoughts on this matter. Regarding ethics, we have rigorously adhered to established guidelines and included comprehensive ethical statements in our paper's Appendix D, aligning with the standards in the blockchain research community. Although we understand your reservation about assessing these ethical aspects, we assure you of our commitment to ethical compliance and are open to further review or clarification to address any concerns. We value your feedback as it reinforces our dedication to maintaining high standards in both the presentation and ethical conduct of our research.

---

> > > ### Author Response · Authors · 2023-11-17
> > >
> > > 6. There are some technical things that I discuss below, but these seems to be more minor in nature. As an example, the set of features degree (3 features) and neighbor (3 features) seem to be highly linearly correlated (as also suggested by their statistics in Table 3). Keeping only of them, i.e., one from the "degree" family, and one from the "neighbor" family, would potentially improve the performance of the networks. I provide a couple of more minor comments below.
> > >
> > >
> > >
> > >
> > >     We greatly appreciate your technical observations and suggestions. Regarding the features of 'degree' and 'neighbor' that you mentioned, we understand your point about their potential linear correlation as suggested in Table 3. We would like to clarify that while these features are related, they represent distinct aspects of our dataset. Neighbors and degrees provide different perspectives: the number of neighbors highlights direct connections, whereas degrees can encapsulate the intensity of these connections, especially when multiple edges exist between two neighbors. To address your suggestion, we have conduct additional Ethereum link prediction to evaluate the impact of retaining only one feature from each of these categories – one from the 'degree' family and one from the 'neighbor' family. The results are as below.
> > >
> > >
> > >
> > >     **Ethereum Address Link Prediction Without Twitter Features: Original vs. Simplified Feature Sets (Original: Complete 8-dimension Structural Features; Simplified: Selected One Feature Each from 'Degree' and 'Neighbor' Families)**
> > >
> > >     | **Method**     | **Feature Set**           | **AUC-ROC** | **Precision** | **Recall** | **F1 Score** |
> > >     |----------------|---------------------------|-------------|---------------|------------|--------------|
> > >     | **Cluster-GCN** | Original                  | 0.82±0.01   | 0.78±0.01     | 0.70±0.01  | 0.74±0.01    |
> > >     |                | Simplified                | 0.79±0.00(↓)| 0.70±0.02(↓)  | 0.78±0.02(↑)| 0.73±0.00(↓)|
> > >     | **GraphSage**   | Original                  | 0.84±0.00   | 0.77±0.01     | 0.77±0.01  | 0.78±0.00    |
> > >     |                | Simplified                | 0.84±0.00(-)| 0.78±0.01(↑)  | 0.75±0.01(↓)| 0.76±0.01(↓)|
> > >     | **APPNP**       | Original                  | 0.82±0.01   | 0.80±0.03     | 0.68±0.02  | 0.74±0.01    |
> > >     |                | Simplified                | 0.80±0.01(↓)| 0.74±0.02(↓)  | 0.73±0.01(↑)| 0.74±0.00(↑)|
> > >     | **TAGCN**       | Original                  | 0.81±0.01   | 0.82±0.01     | 0.64±0.02  | 0.72±0.01    |
> > >     |                | Simplified                | 0.79±0.01(↓)| 0.80±0.03(↓)  | 0.60±0.02(↓)| 0.73±0.00(↑)|
> > >
> > >
> > >
> > >
> > >
> > >     **Ethereum Address Link Prediction With Twitter Features: Original vs. Simplified Feature Sets (Original: Complete 8-dimension Structural Features; Simplified: Selected One Feature Each from 'Degree' and 'Neighbor' Families)**
> > >
> > >     | **Method**     | **Feature Set**           | **AUC-ROC** | **Precision** | **Recall** | **F1 Score** |
> > >     |----------------|---------------------------|-------------|---------------|------------|--------------|
> > >     | **Cluster-GCN** | Original                  | 0.86±0.00   | 0.81±0.00     | 0.75±0.01  | 0.78±0.00    |
> > >     |                | Simplified                | 0.82±0.02(↓)| 0.78±0.02(↓)  | 0.67±0.01(↓)| 0.73±0.01(↓)|
> > >     | **GraphSage**   | Original                  | 0.86±0.00   | 0.81±0.01     | 0.79±0.01  | 0.80±0.00    |
> > >     |                | Simplified                | 0.85±0.00(↓)| 0.80±0.01(↓)  | 0.76±0.02(↓)| 0.78±0.01(↓)|
> > >     | **APPNP**       | Original                  | 0.89±0.02   | 0.83±0.00     | 0.78±0.00  | 0.81±0.00    |
> > >     |                | Simplified                | 0.88±0.01(↓)| 0.81±0.01(↓)  | 0.81±0.01(↓)| 0.81±0.00(-)|
> > >     | **TAGCN**       | Original                  | 0.88±0.00   | 0.85±0.01     | 0.73±0.01  | 0.78±0.00    |
> > >     |                | Simplified                | 0.85±0.01(↓)| 0.84±0.01(↓)  | 0.66±0.01(↓)| 0.74±0.00(↓)|
> > >
> > >
> > >
> > >
> > >     The results from our experiments provided insightful findings. When we limited the features to just one from the degree family and one from the neighbor family, we generally observed a decrease in prediction performance across the two experimental settings, particularly in the predictions involving Twitter features. This trend highlights the significance of utilizing a comprehensive set of features in our model to effectively capture the intricate details and nuances of our dataset. Your suggestion played a crucial role in leading us to this valuable understanding. We deeply appreciate this opportunity to explore these aspects further, as it has been instrumental in refining and strengthening our approach and methodology.

---

> > > > ### Comment · Reviewer_hHyG · 2023-11-20
> > > >
> > > > I thank the authors for their responses. I acknowledge that they clear many of my doubts regarding the contributions of the paper. The authors seem to have addressed all ethical issues regarding sharing the dataset, and while I am not expert in datasets, I am fairly confident that I can remove this flag. Also, I appreciate the additional ablation studies provided in the rebuttal. Although these are not the main focus of the paper, I wonder if the authors would find it useful to include them in their appendix. In any case, that is up to them to decide. Overall, I appreciate that the revised version has only minor revisions that are easily readable.
> > > >
> > > > In sum, I think that this dataset will be useful to blockchain researchers. If ICLR is accepting such interdisciplinary datasets, I am happy to recommend the paper to be accepted in its current status.

---

> ### Author Response · Authors · 2023-11-17
>
> 7. Can you please discuss the weaknesses above?
>
>    Thank you for prompting a discussion on the weaknesses identified in our study. We have addressed each of the weaknesses mentioned earlier in our responses.
>
> 8. Is overs-smoothing a relevant problem in the evaulations and if yes, have you taken steps to address it?
>
>     Thank you for your question regarding over-smoothing in our evaluations. While over-smoothing is an acknowledged concern in deeper Graph Neural Network (GNN) architectures, it was not a primary focus of our current research. Our study primarily revolves around the development and utilization of a novel dataset linking Ethereum blockchain data with Twitter social network data, and exploring the insights this combined data can provide.
>
>     Though it is not our focus, we conduct some experiments or analysis on employing 2 layers, 3 layers, 4 layers, and 5 layers in GNN models to conduct the Ethereum link prediction with and without Twitter features. We choose GraphSage as our backbone. The experiment results are attached below.
>
>     **Experimental Results: GraphSage Model Layer Comparison for Ethereum Link Prediction**
>
>     This table presents the results from experiments investigating the impact of model depth on over-smoothing in GraphSage models. We compare the performance of two, three, four, and five-layer models in conducting Ethereum link prediction, both with and without incorporating Twitter features.
>
>     | **Number of Layers** | **Feature Set**           | **AUC-ROC** | **Precision** | **Recall** | **F1 Score** |
>     |----------------------|---------------------------|-------------|---------------|------------|--------------|
>     | **Two Layers**         | Without Twitter Features  |     0.82±0.01        |      0.79±0.02         |    0.71±0.02        |     0.75±0.01         |
>     |                      | With Twitter Features     |     0.86±0.01        |      0.82±0.01         |       0.72±0.03     |  0.76±0.01             |
>     | **Three Layers**         | Without Twitter Features  |    0.84±0.00         |    0.77±0.01           |     0.77±0.01       |     0.78±0.00         |
>     |                      | With Twitter Features     |     0.86±0.00        |   0.81±0.01            |     0.79±0.01       |      0.80±0.00        |
>     | **Four Layers**         | Without Twitter Features  |   0.76±0.02          |     0.69±0.03           |  0.71±0.01          |      0.70±0.02        |
>     |                      | With Twitter Features     |     0.75±0.03        |    0.70±0.03           |      0.67±0.01      |  0.69±0.01            |
>     | **Five Layers**         | Without Twitter Features  |   0.71±0.02          |     0.65±0.03          |     0.67±0.03       |     0.66±0.03         |
>     |                      | With Twitter Features     |     0.72±0.01         |      0.71±0.03        |    0.53±0.01        |     0.61±0.01          |
>
>
>
>
>     Our experimental results indicate a decline in prediction performance when the number of layers in the model exceeds three. This finding underscores the issue of oversmoothing, particularly evident from the fourth layer onwards in this dataset.
>
>     In conclusion, increasing the number of layers beyond three introduces oversmoothing challenges. We would like to reiterate that this paper primarily focuses on introducing a pioneering dataset that bridges Ethereum and Twitter. Following your suggestion, we plan to delve deeper into the oversmoothing issue in future research using this dataset.
>
> 9. Figure 2: In the Feature Extraction column, we read in the first box "Degree"-> Structural Features. So, here, you mean that look at the degree of the node or that "Degree" is some kind of network like BERT and Deepwalk in the next two boxes of that column? I found this slightly confusing. Also, which structural features are extracted here since ETH addresses don't contain any further information.
>
>    Thank you for your question regarding Figure 2's Feature Extraction column. In this context, the term 'Degree' specifically denotes the degree of a node within the network, rather than referring to a network type like BERT or DeepWalk. For Ethereum addresses, we have extracted a set of 8-dimensional structural features including: degree, in-degree, out-degree, neighbors, in-neighbors, out-neighbors, maximum transaction numbers with neighbors, and average neighbor degree. These features are comprehensively detailed in the Section 4.1.2 and Appendix C.1 of our paper, as space limitations in the figure restricted us from including all the specifics. We hope this clarification resolves your confusion and provides a clearer understanding of our feature extraction methodology.

---

> > ### Author Response · Authors · 2023-11-17
> >
> > 10. Below Figure 2: "The total collection accounts for 30,387 Twitter profiles". I was confused here: why is this lower than the initial number (since, according to the previous lines more data was added) and why are Followers and Friends useful here? Friends are not discussed elsewhere if I am not wrong.
> >
> >     Thank you for highlighting the discrepancy regarding the total number of Twitter profiles mentioned below Figure 2. The figure of 30,387 represents the active Twitter accounts at the time of constructing our graph. This number is lower than the initial count as it accounts for only those profiles that were accessible and active during our data collection process. We have updated this figure to 30,387 in blue color in our revised paper's section 3.1. We appreciate your attention to this detail.
> >
> >
> > 11. and why are Followers and Friends useful here? Friends are not discussed elsewhere if I am not wrong.
> >
> >     We are thankful for the chance to clarify the importance of incorporating 'Followers' and 'Friends' data in our study. These metrics are crucial for evaluating the influence and network span of Twitter accounts. In our research, 'Friends' specifically denotes the accounts followed by a Twitter user. We have meticulously gathered data on the followers and followed accounts of Twitter users linked to Ethereum addresses, playing a vital role in forming a comprehensive and directed Twitter graph. Additionally, the quantity of 'Friends' a Twitter account maintains is indicative of its engagement level, thereby enhancing our analysis with valuable insights into user engagement and interaction trends within the Twitter network. In our updated paper, we have provided detailed explanations about 'Followers' and 'Friends' in Section 3.1, highlighted in blue for easy reference. Thank you once more for this perceptive inquiry.
> >
> > 12. should it be Twitter or X :)?
> >
> >     In response to your query about the correct term - Twitter or X - the data we collected was valid until August 2022, and it referred to Twitter at that time. We appreciate your attention to this detail and confirm that the term 'Twitter' is used accurately in our context.
> >
> > 13. is it 19 billion or 190 billion as of September 23, the market value for Ethereum (page 1)?
> >
> >     Concerning the market value of Ethereum, your observation about the discrepancy is well-founded. As of September 23, the accurate market value was indeed $190 billion, not $19 billion as previously stated. We have amended this figure in the introduction, now highlighted in blue, in our updated paper. We are grateful for your keen attention to detail and have made the appropriate correction in our revised manuscript to accurately reflect this figure.
> >
> > 14. please define abbreviations the first time that they are used, e.g., AUC.
> >
> >     We sincerely appreciate your recommendation to define abbreviations upon their first occurrence. In response to your valuable suggestion, we have revised our manuscript to include explicit definitions for all abbreviations, such as AUC (Area Under the Curve), which we have highlighted in blue in the introduction. This enhancement significantly improves the clarity and accessibility of our paper. We are grateful for your insightful advice!
> >
> > 15. citations
> >
> >     Huang, Shenyang, et al. "Temporal graph benchmark for machine learning on temporal graphs." arXiv preprint arXiv:2307.01026 (2023).
> >
> >
> > Finally, we extend our sincere gratitude for the thorough reviews and valuable feedback provided. We deeply appreciate this opportunity to refine our paper and provide additional insights into our experiments.

---

> ### Author Response · Authors · 2023-11-21
>
> Dear Reviewer hHyG,
>
> We are sincerely thankful for your decision to increase the score of our submission, which greatly affirms the value of our work!
>
> And thank you for your thoughtful and constructive review of our paper. We are gratified to hear that our responses and revisions have effectively addressed your concerns, particularly regarding the ethical considerations of sharing our dataset. Your confidence in removing the flag related to dataset concerns is immensely appreciated.
>
> We also value your appreciation of the additional ablation studies included in our rebuttal. Your suggestion to possibly include these in the appendix is intriguing and something we will thoughtfully consider. It's encouraging to know that these supplementary studies, while not the main focus, have contributed positively to the overall quality of our work.
>
> Your acknowledgment of the readability and coherence of the revised version is especially heartening, as we endeavored to make the paper accessible and clear.
>
> Your expressed confidence in the utility of our dataset for blockchain researchers, coupled with your endorsement for the paper's acceptance at ICLR, is profoundly encouraging. We are thrilled at the opportunity to contribute to the interdisciplinary nexus of blockchain and machine learning research, and we eagerly anticipate the impact our work will have in this vibrant and evolving field.
>
> Once again, thank you for your insightful feedback and support. Your review has been instrumental in refining our paper and ensuring its relevance and clarity.
>
> Warm regards,
>
> All authors

---

### Official Review · Reviewer_AoG7 · 2023-10-26

**Soundness:** 3 good
**Presentation:** 3 good
**Contribution:** 2 fair
**Rating:** 5
**Confidence:** 3

**Summary:**

The paper constructs a new graph dataset which uses Ethereum NFT transaction records together with Twitter social network. With the information from Twitter graph, the authors use BERT and deep walk to extract semantic and topology information, which is used as node feature for node on Ethereum graph. They conduct experiments on link prediction, wash-trading addresses detection, and matching link prediction tasks, demonstrating with the help of twitter features, various kinds of GNN methods can get performance improvements on all these tasks.

**Strengths:**

1. The paper proposes a novel idea to link Twitter social network to Ethereum networks which can provide node features for Ethereum networks.
2. Experiments are conducted thoroughly and prove that with Twitter features, most kinds of models can achieve better performance. Dataset statistics are provided clearly.

**Weaknesses:**

1. Though the title says ‘bridging Ethereum and Twitter’, the proposed dataset is focused on NFT transactions and related Twitter accounts. The author should either prove that with this dataset, GNN would have the ability to find other kinds of matching links (i.e., phish-hack EOA nodes and corresponding Twitter accounts), or modify the title to be more precise.

2. In Section 3.3, the authors claim, ‘we obtain embeddings for all Twitter accounts using the DeepWalk algorithm in the Twitter graph’. Figure 2 also shows structural features given by deep walk. However, as stated in Appendix, only the handcrafted 8 features from C.1 are used.

3. For task 1, the authors only use matched addresses. Since GNN can pass node features via message-passing mechanism, the semantic information is able to spread around. Whether does the semantic information benefit or harm the prediction of non-matched addresses?

4. For task 3 is to predict whether one Ethereum address is connected to one Twitter account. In Appendix K, the authors describe
"For Twitter accounts that do not have a matched Ethereum address, we ... appending an eight-dimensional zero-vector to the existing structural features."
 While in Section 4.4, the negative samples are pick non-existing connections between Ethereum addresses and all their matched Twitter accounts. If the negative samples are $\{e_k=(v_i, v_j)| v_i \in V_\text{eth}, v_j\in V_{\text{twitter with profile}}, e_k\notin E_\text{match}\}$, why are there Twitter accounts with no Ethereum addresses? If the negative samples are $\{e_k=(v_i, v_j)| v_i \in V_\text{eth}, v_j\in V_{\text{twitter}}, e_k\notin E_\text{match}\}$, then does the input contain bias already?

**Questions:**

See the weakness part.

---

> ### Author Response · Authors · 2023-11-17
> **Response to Reviewer AoG7**
>
> Dear Reviewer AoG7,
>
> We express our sincere gratitude for your thorough and constructive review of our paper. Your recognition of the novelty in linking the Twitter social network with Ethereum networks, and the clear presentation of our dataset statistics, is greatly appreciated. We are also thankful for your acknowledgment of our comprehensive experimental approach, demonstrating the enhanced performance of various GNN methods with the integration of Twitter features. Your positive feedback on the soundness, presentation, and contribution of our work is encouraging and validates the efforts we have put into this research. In response to the weaknesses you have identified, we offer detailed, point-by-point explanations below.
>
>
> 1. Though the title says ‘bridging Ethereum and Twitter’, the proposed dataset is focused on NFT transactions and related Twitter accounts. The author should either prove that with this dataset, GNN would have the ability to find other kinds of matching links (i.e., phish-hack EOA nodes and corresponding Twitter accounts), or modify the title to be more precise.
>
>     Thank you for your thoughtful feedback on our paper titled “Bridging Ethereum and Twitter.” We understand your concerns about the focus on NFT transactions and corresponding Twitter accounts. This title was chosen to showcase our pioneering research of the integration between Ethereum and Twitter data, considering that the NFT data was extracted Ethereum transactions. Although we recognize the benefits of a more specific title, the ICLR 2024 guidelines limit modifications to the title and abstract during the rebuttal phase. Should our paper be accepted, we will revise the title to “ETGraph: A Pioneering Dataset Bridging Ethereum NFTs and Twitter” for the camera-ready version, in line with ICLR’s policy for minor adjustments. We appreciate your valuable suggestion!
>
> 2. In Section 3.3, the authors claim, ‘we obtain embeddings for all Twitter accounts using the DeepWalk algorithm in the Twitter graph’. Figure 2 also shows structural features given by deep walk. However, as stated in Appendix, only the handcrafted 8 features from C.1 are used.
>
>     We are grateful for your attention to detail in highlighting the discrepancy between Section 3.3 and the Appendix C.1 of our paper. You are correct in noting that we initially neglected to include the details regarding the use of the DeepWalk algorithm for obtaining Twitter features in the Appendix C.1. We have addressed this oversight and updated Appendix C.1 to correctly depict the use of DeepWalk, now highlighted in blue, ensuring consistency with the methodology outlined in Section 3.3 and Figure 2. Thank you for bringing this to our attention, allowing us to improve the clarity and consistency of our paper!

---

> ### Author Response · Authors · 2023-11-17
>
> 3. For task 1, the authors only use matched addresses. Since GNN can pass node features via message-passing mechanism, the semantic information is able to spread around. Whether does the semantic information benefit or harm the prediction of non-matched addresses?
>
>     We greatly appreciate your insightful query regarding the use of matched addresses in Task 1 and the potential effect of semantic information on the prediction of non-matched addresses! In response, we conducted an additional experiment focusing on Ethereum link prediction among non-matched nodes using multiple GNN models, where the training, validation, and test sets were devoid of any matching link addresses. The results are attached below:
>
>     **Results for non-matching Ethereum addresses' Link Prediction**
>     *(Without: without Twitter semantic information; With: with Twitter semantic information)*
>
>     | **Method**    | **Experimental Setting**                  | **AUC-ROC** | **Precision** | **Recall** | **F1 Score** |
>     |---------------|---------------------------------------|-------------|---------------|------------|--------------|
>     | GCN           | Without         | 0.84±0.01   | 0.76±0.01     | 0.75±0.01  | 0.75±0.00    |
>     |               | With             | 0.84±0.00(↑)| 0.76±0.00(↑)  | 0.75±0.02(↓)| 0.76±0.01(↑)|
>     | GAT           | Without         | 0.78±0.02   | 0.72±0.01     | 0.68±0.03  | 0.70±0.02    |
>     |               | With            | 0.85±0.01(↑)| 0.76±0.00(↑)  | 0.79±0.02(↑)| 0.77±0.01(↑)|
>     | APPNP         | Without         | 0.84±0.01   | 0.80±0.00     | 0.70±0.01  | 0.74±0.00    |
>     |               | With            | 0.87±0.00(↑)| 0.79±0.00(↓)  | 0.77±0.00(↑)| 0.78±0.00(↑)|
>     | Cluster-GCN   | Without         | 0.88±0.00   | 0.81±0.00     | 0.78±0.01  | 0.80±0.01    |
>     |               | With              | 0.90±0.00(↑)| 0.82±0.00(↑)  | 0.82±0.01(↑)| 0.82±0.00(↑)|
>     | TAGCN         | Without         | 0.84±0.00   | 0.81±0.00     | 0.71±0.00  | 0.76±0.00    |
>     |               | With           | 0.85±0.00(↑)| 0.80±0.00(↓)  | 0.74±0.01(↑)| 0.77±0.00(↑)|
>
>     *Note: '↑' indicates an increase, '↓' indicates a decrease.*
>
>     The results of this experiment indicate that in most cases, the semantic information indeed benefits the prediction of non-matched addresses. This outcome not only corroborates the capability of GNNs to disseminate node features through the message-passing mechanism but also underscores the utility of our dataset in enhancing Ethereum link predictions, even in the absence of direct links to Twitter. Your feedback has been instrumental in guiding this valuable extension of our research. We have added this experiment to our revised Appendix I.4, highlighting its subtitle in blue color. Thank you again for your insight!

---

> ### Author Response · Authors · 2023-11-17
>
> 4. For task 3 is to predict whether one Ethereum address is connected to one Twitter account. In Appendix K, the authors describe "For Twitter accounts that do not have a matched Ethereum address, ... why are there Twitter accounts with no Ethereum addresses? If the negative samples are ... then does the input contain bias already?
>
>     Thank you for your observations on Task 3 of our study. We appreciate this opportunity to clarify our methodology:
>
>     **Objective of Task 3**: Our objective in Task 3 is to accurately determine if a specific Ethereum address (A) is connected to its real matching Twitter account (B). All the edges we analyze are actual matching links between Ethereum and Twitter accounts we have collected.
>
>     **Unmatched Twitter Accounts**: We try our best to collect the graph as large as possible to collect more informative data from both Twitter and Ethereum, and collect matching link information as much as possible. Then, on both of them, we conduct GNN to calculate node representations based on our collected sufficient data. Though many Twitter accounts do not have matching link information, they are important because they contribute to the node representation calculation by utilizing the GNN to facilitate the message passing between Twitter nodes, in order to enhance the prediction of matching links.
>
>     For Twitter accounts that do not have a corresponding Ethereum address, our methodology involves appending an eight-dimensional zero-vector to their respective feature sets. This procedure is vital for preserving the dimensional uniformity among all nodes within the most extensive connected Twitter graph, which is a critical factor for the optimal performance of GNN. It is important to highlight that, although these zero-vector features do not represent active attributes, they play a crucial role in upholding the structural integrity necessary for GNN analysis.
>
>     **Negative Sample Selection**: In our study, we carefully curated negative samples comprising hypothetical, non-existent connections between Ethereum addresses and Twitter accounts. This methodology was intentionally adopted to enhance our model's ability to discern between genuine and artificial connections, thereby improving its overall predictive precision. It's important to note that these negative samples were meticulously selected from the pool of nodes involved in actual matching links within our dataset. To address concerns about potential biases, especially between matching and non-matching Twitter accounts, we consciously avoided choosing Twitter nodes that do not have corresponding matches in Ethereum for our negative samples. This decision aligns with the second scenario you outlined, ensuring that our experimental design remains balanced and free from unintended biases.
>
>     **Furthermore, in response to your concerns, we have conducted a supplementary experiment to investigate whether selecting non-matching Twitter accounts introduces bias in our analysis. The experiment results are attached below.**

---

> ### Author Response · Authors · 2023-11-17
>
> **Comparative Results for Task 3: Ethereum-Twitter Matching Link Prediction**
>
> This table showcases the findings from our experiments for Task 3, centered on predicting the links between Ethereum addresses and Twitter accounts. In our comparative analysis, we evaluated the performance of two distinct setups: our original configuration, which employs matching Twitter accounts, and an alternative approach where we utilized non-matching Twitter accounts as negative samples. This comparison aims to provide a clearer understanding of the impact of negative sample selection on the accuracy and reliability of our predictions.
>
>
>
> | **Model**     | **Negative Samples**        | **AUC-ROC** | **Precision** | **F1 Score** | **Accuracy** |
> |---------------|------------------------|-------------|---------------|------------|--------------|
> | **GCN**       | Original | 0.71±0.01   | 0.64±0.01     | 0.65±0.03  | 0.71±0.01    |
> |               | Non-matching| 0.94±0.00(↑)| 0.91±0.00(↑)  | 0.82±0.01(↑)| 0.84±0.01(↑)|
> | **Cluster-GCN** |Original  | 0.67±0.01   | 0.65±0.01     | 0.66±0.01  | 0.65±0.01    |
> |               | Non-matching | 1.00±0.00(↑)| 1.00±0.00(↑)  | 0.99±0.01(↑)| 0.99±0.01(↑)|
> | **GraphSage**   | Original| 0.67±0.02   | 0.63±0.02     | 0.71±0.02  | 0.67±0.01    |
> |               | Non-matching| 1.00±0.00(↑)| 1.00±0.00(↑)  | 1.00±0.00(↑)| 1.00±0.00(↑)|
>
>
> *Note: '↑' indicates an increase, '↓' indicates a decrease, and '-' indicates no significant change in performance.*
>
>
> The results of our experiment offer a clear indication that selecting non-matching Twitter accounts as negative samples significantly influences prediction accuracy, leading to near-perfect scores close to 100%. This high level of accuracy can be attributed to the fact that Twitter accounts without corresponding Ethereum addresses lack distinct Twitter profiles in our dataset. Instead, we use an eight-dimensional zero vector as a placeholder for them. Such uniformity in features makes these accounts easily identifiable during the training of Graph Neural Networks (GNNs), leading to an unusually high prediction accuracy, which is a bias as you mentioned. This finding corroborates your concern regarding the introduction of bias in our analysis when non-matching Twitter accounts are used as negative samples.
>
>
>
>
>
>
>
> Last but not least, we sincerely thank you for your thoughtful and detailed review of our work. Your insightful feedback has been instrumental in refining our approach and methodology, particularly in the nuanced aspects of handling unmatched Twitter accounts and the selection of negative samples for our GNN analysis. Your questions and observations have not only helped in clarifying these key areas but have also contributed significantly to enhancing the overall quality and clarity of our research. We are truly grateful for the time and effort you have dedicated to reviewing our paper.

---

> > ### Author Response · Authors · 2023-11-21
> > **Looking forward to your reply**
> >
> > Dear Reviewer AoG7,
> >
> > We would like to express our deep gratitude for your comprehensive and insightful review of our paper. In response to the points you raised, we have provided detailed, point-by-point explanations and additional experiments to address your concerns thoroughly.
> >
> > Given the importance of these updates and the potential impact they could have on the evaluation of our work, we kindly request that you review our rebuttal at your earliest convenience. Your further insights and feedback would be invaluable in ensuring that our paper meets the highest standards of quality and rigor. We eagerly await your response and are ready to engage in further discussion to refine our work as needed.
> >
> > Thank you once again for your thoughtful review and the time you have invested in our paper. Your expertise and guidance are crucial in helping us achieve our goal of contributing meaningful research to the blockchain and machine learning research community.
> >
> >
> > Best regards,
> >
> > All authors

---

> > > ### Author Response · Authors · 2023-11-22
> > > **Looking forward to your reply**
> > >
> > > Dear Reviewer AoG7,
> > >
> > > Thank you for the time and effort you've invested in reviewing our work and providing insightful feedback. In response to the points you raised, we have provided detailed, point-by-point explanations and additional experiments to address your concerns thoroughly. We recognize your busy schedule, but with the discussion deadline nearing, we kindly request your review of our responses. We eagerly anticipate your reply and are more than willing to clarify any additional questions you may have.
> > >
> > > Best regards,
> > >
> > > All authors

---

> > > > ### Author Response · Authors · 2023-11-23
> > > > **Request for Review and Feedback as Discussion Deadline Approaches**
> > > >
> > > > Dear Reviewer AoG7,
> > > >
> > > > Thanks a lot for your time in reviewing and insightful comments. We sincerely understand you’re busy. But since the discussion due is approaching, would you mind checking the response to confirm where you have any further questions?
> > > >
> > > > We are looking forward to your reply and happy to answer your further questions.
> > > >
> > > >
> > > > Best regards,
> > > >
> > > > All authors

---

> > > > > ### Author Response · Authors · 2023-11-23
> > > > > **Gentle Reminder: Awaiting Your Review of Our Submitted Response**
> > > > >
> > > > > Dear Reviewer AoG7,
> > > > >
> > > > > With the discussion deadline just around the corner, only two hours away, we wish to briefly summarize our responses to your primary concerns and are hopeful for your reply!
> > > > >
> > > > > **Q1: Whether does the semantic information benefit or harm the prediction of non-matched addresses?**
> > > > >
> > > > > We conducted an additional experiment focusing on Ethereum link prediction among non-matched Ethereum addresses using multiple GNN models, where the training, validation, and test sets were devoid of any matching link addresses. The results of this experiment indicate that in most cases, the semantic information indeed benefits the prediction of non-matched Ethereum addresses. This outcome not only corroborates the capability of GNNs to disseminate node features through the message-passing mechanism but also underscores the utility of our dataset in enhancing Ethereum link predictions, even in the absence of direct links to Twitter.
> > > > >
> > > > > **Q2: While in Section 4.4, the negative samples are pick non-existing connections between Ethereum addresses and all their matched Twitter accounts. Why are there Twitter accounts with no Ethereum addresses? And if the negative samples include Twitter accounts without profiles, will it introduce bias?**
> > > > >
> > > > > a) Why are there Twitter accounts with no Ethereum addresses?
> > > > >
> > > > > We try our best to collect the graph as large as possible to collect more informative data from both Twitter and Ethereum, and collect matching link information as much as possible. Then, on both of them, we conduct GNN to calculate node representations based on our collected sufficient data. Though many Twitter accounts do not have matching link information, they are important because they contribute to the node representation calculation by utilizing the GNN to facilitate the message passing between Twitter nodes, in order to enhance the prediction of matching links.
> > > > >
> > > > > b) And if the negative samples include Twitter accounts without profiles, will it introduce bias?
> > > > >
> > > > > In our study, we carefully curated negative samples comprising hypothetical, non-existent connections between Ethereum addresses and Twitter accounts. This methodology was intentionally adopted to enhance our model's ability to discern between genuine and artificial connections, thereby improving its overall predictive precision. It's important to note that these negative samples were meticulously selected from the pool of nodes involved in actual matching links within our dataset.
> > > > >
> > > > > In response, we have conducted a supplementary experiment to investigate whether selecting non-matching Twitter accounts introduces bias in our analysis. The results of our experiment offer a clear indication that selecting non-matching Twitter accounts as negative samples significantly influences prediction accuracy, leading to near-perfect scores close to 100%. This high level of accuracy can be attributed to the fact that Twitter accounts without corresponding Ethereum addresses lack distinct Twitter profiles in our dataset. Instead, we use an eight-dimensional zero vector as a placeholder for them. Such uniformity in features makes these accounts easily identifiable during the training of GNN, leading to an unusually high prediction accuracy, which is a bias as you mentioned.
> > > > >
> > > > > Thank you for your attention and time dedicated to reviewing our responses. We deeply appreciate your invaluable input and guidance throughout this process. Your prompt feedback, especially considering the tight timeline, is greatly appreciated and will be instrumental in the successful completion of our discussion. Thank you once again for your expertise and support.
> > > > >
> > > > > Happy Thanksgiving!
> > > > >
> > > > >
> > > > > Best regards,
> > > > >
> > > > > All authors

---

### Official Review · Reviewer_SzMQ · 2023-10-27

**Soundness:** 3 good
**Presentation:** 3 good
**Contribution:** 3 good
**Rating:** 6
**Confidence:** 5

**Summary:**

The paper delves into the analysis of Ethereum activities, specifically examining the influence of Twitter data on these activities. Through experiments, the authors compare the outcomes of Ethereum link predictions and wash-trading address detections with and without the integration of Twitter features. Their findings suggest that incorporating Twitter data can significantly enhance the performance of various graph-based models in these tasks. The research leverages advanced computational tools and the ETGraph dataset, emphasizing the potential synergy between blockchain activities and social media data.

**Strengths:**

The paper provides a new dataset for graph representation with Ethereum blockchain and Twitter data. As there are already existing datasets for the separated dataset, the combined dataset with opensea may be considered a contribution. The authors experimentally proved that combining the dataset improved the performance of Ethereum link prediction and wash trading detection using various existing methods.
The novel dataset could be used in both the blockchain and ML communities to compete with SOTA for graph learning algorithms.

**Weaknesses:**

As the paper suggests a new dataset, it is acknowledged that the technical novelty is not strong enough. But as the authors provide a webpage and github page to easily use the dataset, it may contribute to the ML community.

**Questions:**

No questions.

---

> ### Author Response · Authors · 2023-11-17
> **Response to Reviewer SzMQ**
>
> Dear Reviewer SzMQ,
>
> We are sincerely grateful for your recognition of our contribution, particularly the introduction of our novel graph dataset, ETGraph. This dataset, which uniquely combines Ethereum and Twitter data, holds promising potential for impactful applications in both the blockchain and machine learning communities. In response to the weaknesses you have highlighted, we provide detailed explanations below.
>
> 1. As the paper suggests a new dataset, it is acknowledged that the technical novelty is not strong enough.
>
>     Thank you for bringing up this point. We acknowledge your concerns about the technical novelty of our work. Nonetheless, it's important to emphasize that the development of our dataset transcends simple data aggregation. Our process entailed intricate data collection from Ethereum and Twitter, followed by sophisticated integration and preprocessing tasks, such as graph construction and feature engineering. These elements collectively enhance the technical complexity of our project, setting it apart in the landscape of machine learning research.
>
>
> 2. But as the authors provide a webpage and github page to easily use the dataset, it may contribute to the ML community.
>
>    Thank you for acknowledging our contribution to the ML community. Our decision to provide easy access to the dataset through a well-organized website and GitHub repository is a conscious effort to support and stimulate further research. We believe that this accessibility is a vital contribution to the machine learning community, facilitating broader use and exploration of ETGraph.
>
> Once again, we extend our gratitude for your thoughtful review and the valuable insights you have provided. Your feedback has not only strengthened our research but also serves as a source of motivation as we continue to pursue innovative endeavors within the machine learning community.

---

> > ### Author Response · Authors · 2023-11-21
> > **Looking forward to your reply**
> >
> > Dear Reviewer SzMQ,
> >
> > We deeply appreciate your time and effort in reviewing our manuscript and recognizing the potential of our novel graph dataset, ETGraph. Your insights and observations have been invaluable in guiding us to refine and improve our work. We understand your concern about the technical novelty of the dataset. **While the dataset itself may not introduce new technical methodologies, its unique combination of Ethereum and Twitter data presents a new frontier for exploratory research in both blockchain and machine learning fields.**
> >
> > We would appreciate it if you could provide any feedback and raise the score. Thanks for your time and effort!
> >
> > Best regards,
> >
> > All authors

---

### Author Response · Authors · 2023-11-20
**Looking forward to your reply**

Dear Reviewers,

We have addressed your concerns and revised the paper. We would appreciate it if you could provide any feedback and raise the score if your questions are solved. Thanks for your time and effort!


Best regards,

All authors

---

### Author Response · Authors · 2023-11-21
**Summary of Our Rebuttal**

Dear All Reviewers,

We extend our heartfelt gratitude to all reviewers for dedicating their time to evaluate our work. It is encouraging to see the acknowledgment of our unique contribution, which is **the introduction of a novel graph dataset combining Ethereum blockchain and Twitter data** (as noted by Reviewer #SzMQ, #AoG7, and #hHyG). **The novelty of our idea has been appreciated** (by Reviewers #SzMQ, #AoG7, and #hHyG), along with **the thoroughness of our experiments** (Reviewer #AoG7). We are also gratified by the recognition of our work's **high reproducibility** (Reviewer #hHyG) and the evaluations demonstrating a generally **enhanced performance across a broad spectrum of benchmark GNN models** (as highlighted by Reviewers #SzMQ, #AoG7, and #hHyG).

Here, we provide an overview of responses for main questions proposed by reviewers for convenient reading.

**Q1: Whether does the semantic information benefit or harm the prediction of non-matched addresses** (Reviewer #AoG7)

We conducted an additional experiment focusing on Ethereum link prediction among non-matched Ethereum addresses using multiple GNN models, where the training, validation, and test sets were devoid of any matching link addresses. The results of this experiment indicate that in most cases, the **semantic information indeed benefits the prediction of non-matched Ethereum addresses.** This outcome not only corroborates the capability of GNNs to disseminate node features through the message-passing mechanism but also underscores the utility of our dataset in enhancing Ethereum link predictions, even in the absence of direct links to Twitter.

**Q2: While in Section 4.4, the negative samples are pick non-existing connections between Ethereum addresses and all their matched Twitter accounts. Why are there Twitter accounts with no Ethereum addresses? And if the negative samples include Twitter accounts without profiles, will it introduce bias?** (Reviewer #AoG7)

**a) Why are there Twitter accounts with no Ethereum addresses?**

We try our best to collect the graph as large as possible to collect more informative data from both Twitter and Ethereum, and collect matching link information as much as possible. Then, on both of them, we conduct GNN to calculate node representations based on our collected sufficient data. Though many Twitter accounts do not have matching link information, they are important because **they contribute to the node representation calculation by utilizing the GNN to facilitate the message passing between Twitter nodes**, in order to enhance the prediction of matching links.

**b) And if the negative samples include Twitter accounts without profiles, will it introduce bias?**

In our study, we carefully curated negative samples comprising hypothetical, non-existent connections between Ethereum addresses and Twitter accounts. This methodology was intentionally adopted to enhance our model's ability to discern between genuine and artificial connections, thereby improving its overall predictive precision. **It's important to note that these negative samples were meticulously selected from the pool of nodes involved in actual matching links within our dataset.**

In response, we have conducted a supplementary experiment to investigate whether selecting non-matching Twitter accounts introduces bias in our analysis. The results of our experiment offer a clear indication that selecting non-matching Twitter accounts as negative samples significantly influences prediction accuracy, **leading to near-perfect scores close to 100%.** This high level of accuracy can be attributed to the fact that Twitter accounts without corresponding Ethereum addresses lack distinct Twitter profiles in our dataset. Instead, we use an eight-dimensional zero vector as a placeholder for them. **Such uniformity in features makes these accounts easily identifiable during the training of GNN, leading to an unusually high prediction accuracy, which is a bias as Reviewer #AoG7 mentioned.**

**Q3: Since there are only 3 matched addresses in the dataset, why should we anticipate improved performance with this dataset? The results are, thus, questionable to me regarding Q2.** (Reviewer #hHyG)

In response to the reviewer AoG7's in Q1, we discovered that incorporating Twitter data enhances the link prediction for non-matching Ethereum addresses as well. This improvement is largely attributed to the message-passing mechanism inherent in GNN. Through this mechanism, semantic information from the few matched addresses can disseminate throughout the network, particularly influencing nodes directly connected to these matched addresses. Therefore, despite the small number of direct matches, **the indirect influence of these connections via semantic information propagation would significantly contribute to the overall performance enhancement observed in our results.**

---

> ### Author Response · Authors · 2023-11-21
>
> **Q4: As an example, the set of features degree (3 features) and neighbor (3 features) seem to be highly linearly correlated (as also suggested by their statistics in Table 3). Keeping only of them, i.e., one from the "degree" family, and one from the "neighbor" family, would potentially improve the performance of the networks.** (Reviewer #hHyG)
>
> We have conducted additional Ethereum link prediction to evaluate the impact of retaining only one feature from each of these categories – one from the 'degree' family and one from the 'neighbor' family.
>
> The results from our experiments provided insightful findings. When we limited the features to just one from the degree family and one from the neighbor family, **we generally observed a decrease in prediction performance across the two experimental settings, particularly in the predictions involving Twitter features.** This trend highlights the significance of utilizing a comprehensive set of features in our model to effectively capture the intricate details and nuances of our dataset.
>
>
> **Q5: Is overs-smoothing a relevant problem in the evaulations and if yes, have you taken steps to address it?** (Reviewer #hHyG)
>
> Though it is not our focus, we conduct some experiments or analysis on employing 2 layers, 3 layers, 4 layers, and 5 layers in GNN models to conduct the Ethereum link prediction with and without Twitter features. We choose GraphSage as our backbone.
>
> Our experimental results indicate a decline in prediction performance when the number of layers in the model exceeds three. **This finding underscores the issue of oversmoothing, particularly evident from the fourth layer onwards in this dataset.**

---

### Meta-Review · Area_Chair_kQba · 2023-12-11

**Metareview:**

The paper introduces a dataset that links Ethereum transaction networks (2 million nodes and 30 million edges) and Twitter following data (1 million nodes and 3 million edges), bonding 30,667 Ethereum addresses with verified Twitter accounts sourced from OpenSea. Through a statistical analysis, the authors study the differences between Ethereum addresses on Twitter and ones not on twitter.

The main contribution and strength of the paper is the generation of the dataset itself, which is within ICLR scope as per the call for papers. Reviewers have raised concerns regarding presentation and methodology used for structure/link prediction that the authors have addressed during the rebuttal phase. Their additional results and descriptions of their approach should appear in the camera ready version.

**Justification For Why Not Higher Score:**

Reviewers remained a bit lukewarm, as the main contribution is a dataset.

**Justification For Why Not Lower Score:**

The dataset itself is interesting.

---

### Decision · Program_Chairs · 2024-01-16

Accept (poster)